# Learning Versatile Neural Architectures by Propagating Network Codes

**Mingyu Ding[1], Yuqi Huo[2], Haoyu Lu[2], Linjie Yang[3], Zhe Wang[4], Zhiwu Lu[2], Jingdong Wang[5], Ping Luo[1]**
[1]University of Hong Kong,    [2]Gaoling School of Artificial Intelligence, Renmin University of China,
[3]ByteDance Inc.,    [4]SenseTime Research,    [5]Baidu
mingyuding@hku.hk   wangjingdong@outlook.com   pluo@cs.hku.hk

## Abstract

This work explores how to design a single neural network capable of adapting to multiple heterogeneous vision tasks, such as image segmentation, 3D detection, and video recognition. This goal is challenging because both network architecture search (NAS) spaces and methods in different tasks are inconsistent. We solve this challenge from both sides. We first introduce a unified design space for multiple tasks and build a multitask NAS benchmark (NAS-Bench-MR) on many widely used datasets, including ImageNet, Cityscapes, KITTI, and HMDB51. We further propose Network Coding Propagation (NCP), which back-propagates gradients of neural predictors to directly update architecture codes along the desired gradient directions to solve various tasks. In this way, optimal architecture configurations can be found by NCP in our large search space in seconds.

Unlike prior arts of NAS that typically focus on a single task, NCP has several unique benefits. (1) NCP transforms architecture optimization from data-driven to architecture-driven, enabling joint search an architecture among multitasks with different data distributions. (2) NCP learns from network codes but not original data, enabling it to update the architecture efficiently across datasets. (3) In addition to our NAS-Bench-MR, NCP performs well on other NAS benchmarks, such as NAS-Bench-201. (4) Thorough studies of NCP on inter-, cross-, and intra-tasks highlight the importance of cross-task neural architecture design, *i.e.*, multitask neural architectures and architecture transferring between different tasks. Code is available at github.com/dingmyu/NCP [1].

## 1 Introduction

Designing a single neural network architecture that adapts to multiple different tasks is challenging. This is because different tasks, such as image segmentation in Cityscapes (Cordts et al., 2016) and video recognition in HMDB51 (Kuehne et al., 2011), have different data distributions and require different granularity of feature representations. For example, although the manually designed networks ResNet (He et al., 2016) and HRNet (Wang et al., 2020a) work well on certain tasks such as image classification on ImageNet, they deteriorate in the other tasks. Intuitively, manually designing a single neural architecture that is applicable in all these tasks is difficult.

Recently, neural architecture search (NAS) has achieved great success in searching network architectures automatically. However, existing NAS methods (Wu et al., 2019; Liang et al., 2019; Liu et al., 2019b; Xie et al., 2018; Cai et al., 2020; Yu et al., 2020; Shaw et al., 2019b;a) typically search on a single task. Though works (Ding et al., 2021; Duan et al., 2021; Zamir et al., 2018) designed NAS algorithms or datasets that can be used for multiple tasks, they still search different architectures for different tasks, indicating that the costly searching procedure needs to be repeated many times. The problem of learning versatile neural architectures capable of adapting to multiple different tasks remains unsolved, *i.e.*, searching a multitask architecture or transferring architectures between different tasks. In principle, it faces the challenge of task and dataset inconsistency.

Different tasks may require different granularity of feature representations, *e.g.*, the segmentation task requires more multi-scale features and low-level representations than classification. The key to solving task inconsistency is to design a unified architecture search space for multiple tasks. In contrast to most previous works that simply extend search space designed for image classification

---

[1]Project page: https://network-propagation.github.io

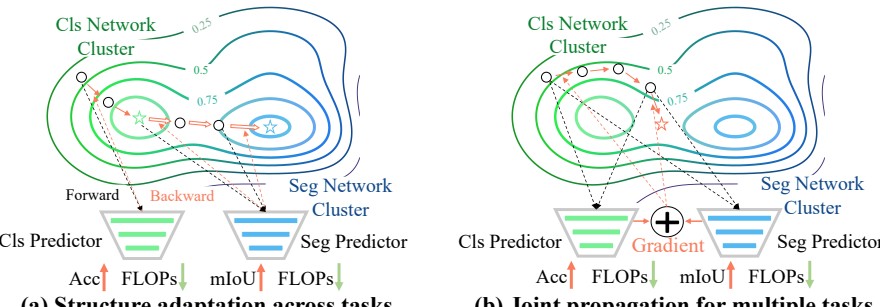

**(a) Structure adaptation across tasks**     **(b) Joint propagation for multiple tasks**

Figure 1: NCP optimizes and propagates the network code in an architecture coding space to achieve the target constraints with back-propagation on the neural predictors. (a) NCP searches for an optimal structure on classification, then adapts it to segmentation through the segmentation predictor. (b) joint propagation for two tasks by accumulating gradients of two predictors.

for other tasks and build NAS benchmarks (Dong & Yang, 2020; Ying et al., 2019; Siems et al., 2020) on small datasets (*e.g.*, CIFAR10, ImageNet-16) with unrealistic settings, we design a multi-resolution network space and build a multi-task practical NAS benchmark (NAS-Bench-MR) on four challenging datasets including ImageNet-224 (Deng et al., 2009), Cityscapes (Cordts et al., 2016), KITTI (Geiger et al., 2012), and HMDB51 (Kuehne et al., 2011). Inspired by HRNet (Wang et al., 2020a), our network space is a multi-branch multi-resolution space that naturally contains various granularities of representations for different tasks, *e.g.*, high-resolution features (Wang et al., 2020a) for segmentation while low-resolution ones for classification. NAS-Bench-MR closes the gap between existing benchmarks and NAS in multi-task and real-world scenarios. It serves as an important contribution of this work to facilitate future cross-task NAS research.

To solve the challenge of dataset inconsistency, in this work, we propose a novel predictor-based NAS algorithm, termed Network Coding Propagation (NCP), for finding versatile and task-transferable architectures. NCP transforms data-oriented optimization into architecture-oriented by learning to traverse the search space. It works as follows. We formulate all network hyperparameters into a coding space by representing each architecture hyper-parameter as a code, *e.g.*, '3,2,64' denotes there are 3 blocks and each block contains 2 residual blocks with a channel width of 64. We then learn neural predictors to build the mapping between the network coding and its evaluation metrics (*e.g.*, Acc, mIoU, FLOPs) for each task. By setting high-desired accuracy of each task and FLOPs as the target, we back-propagate gradients of the learned predictor to directly update values of network codes to achieve the target. In this way, good architectures can be found in several forward-backward iterations in seconds, as shown in Fig. 1.

NCP has several appealing benefits: (1) NCP addresses the data mismatch problem in multi-task learning by learning from network coding but not original data. (2) NCP works in large spaces in seconds by back-propagating the neural predictor and traversing the search space along the gradient direction. (3) NCP can use multiple neural predictors for architecture transferring across tasks, as shown in Fig. 1(a), it adapts an architecture to a new task with only a few iterations. (4) In NCP, the multi-task learning objective is transformed to gradient accumulation across multiple predictors, as shown in Fig. 1(b), making NCP naturally applicable to various even conflicting objectives, such as multi-task structure optimization, architecture transferring across tasks, and accuracy-efficiency trade-off for specific computational budgets.

Our main contributions are three-fold. (1) We propose Network Coding Propagation (NCP), which back-propagates the gradients of neural predictors to directly update architecture codes along desired gradient directions for various objectives. (2) We build NAS-Bench-MR on four challenging datasets under practical training settings for learning task-transferable architectures. We believe it will facilitate future NAS research, especially for multi-task NAS and architecture transferring across tasks. (3) Extensive studies on inter-, intra-, cross-task generalizability show the effectiveness of NCP in finding versatile and transferable architectures among different even conflicting objectives and tasks.

## 2 RELATED WORK

**Neural Architecture Search Spaces and Benchmarks.** Most existing NAS spaces (Jin et al., 2019; Xu et al., 2019; Wu et al., 2019; Cai et al., 2019; Xie et al., 2018; Stamoulis et al., 2019;

Table 1: Comparisons among five NAS benchmarks. Existing benchmarks are either built on small datasets for image classification, or trained with a single simplified setting. In contrast, our NAS-Bench-MR is built on four widely-used visual recognition tasks and various realistic settings. The architectures in our NAS-Bench-MR are trained following the common practices in real-world scenarios, *e.g.*, $512 \times 1024$ and 500 epochs on the CityScapes dataset (Cordts et al., 2016). It takes about 400,000 GPU hours to build our benchmark using Nvidia V100 GPUs.

| Benchmarks | Datasets | Tasks | Scales | Epochs | Input Sizes | Settings per Task |
|---|---|---|---|---|---|---|
| NAS-Bench-101 (Ying et al., 2019) | Cifar-10 | 1 | $10^8$ | 4/12/36/108 | $32 \times 32$ | different training epochs |
| NAS-Bench-201 (Dong & Yang, 2020) | ImageNet-16 | 1 | $10^4$ | 200 | $16 \times 16$ | single setting |
| NAS-Bench-301 (Siems et al., 2020) | Cifar-10 | 1 | $10^{18}$ | 100 | $32 \times 32$ | single setting |
| TransNAS-Bench-101 (Duan et al., 2021) | Taskonomy | 7 | $10^3$ | $\leq 30$ | $256 \times 256$ | single setting |
| **NAS-Bench-MR (Ours)** | ImageNet, Cityscapes, KITTI, HMDB51 | 4 | $10^{23}$ | $\geq 100$ | $224 \times 224$ $512 \times 1024$ | different image sizes, data scale, number of classes, epochs, pretraining |

Mei et al., 2020; Guo et al., 2020; Dai et al., 2020) are designed for image classification using either a single-branch structure with a group of candidate operators in each layer or a repeated cell structure, *e.g.*, Darts-based (Liu et al., 2019b) and MobileNet-based (Sandler et al., 2018) search spaces. Based on these spaces, several NAS benchmarks (Ying et al., 2019; Dong & Yang, 2020; Siems et al., 2020; Duan et al., 2021) have been proposed to pre-evaluate the architectures. However, the above search spaces and benchmarks are built either on proxy settings or small datasets, such as Cifar-10 and ImageNet-16 ($16 \times 16$), which is less suitable for other tasks that rely on multi-scale information. For those tasks, some search space are explored in segmentation (Shaw et al., 2019a; Liu et al., 2019a; Nekrasov et al., 2019; Lin et al., 2020; Chen et al., 2018) and object detection (Chen et al., 2019; Ghiasi et al., 2019; Du et al., 2020; Wang et al., 2020b) by introducing feature aggregation heads (*e.g.*, ASPP (Chen et al., 2017)) for multi-scale information. Nevertheless, the whole network is still organized in a chain-like single branch manner, resulting in sub-optimal performance. Another relevant work to ours is NAS-Bench-NLP Klyuchnikov et al. (2020), which constructs a benchmark with 14k trained architectures in search space of recurrent neural networks on two language modeling datasets.

Compared to previous spaces, our multi-resolution search space, including searchable numbers of resolutions/blocks/channels, is naturally designed for multiple vision tasks as it contains various granularities of feature representations. Based on our search space, we build NAS-Bench-MR for various vision tasks, including classification, segmentation, 3D detection, and video recognition. Detailed comparisons of NAS benchmarks can be found in Tab. 1.

**Neural Architecture Search Methods.** Generally, NAS trains numerous candidate architectures from a search space and evaluates their performance to find the optimal architecture, which is costly. To reduce training costs, weight-sharing NAS methods (Liu et al., 2019b; Jin et al., 2019; Xu et al., 2019; Cai et al., 2019; Guo et al., 2020; Li & Talwalkar, 2020) are proposed to jointly train a large number of candidate networks within a super-network. Different searching strategies are employed within this framework such as reinforcement learning (Pham et al., 2018), importance factor learning (Liu et al., 2019b; Cai et al., 2019; Stamoulis et al., 2019; Xu et al., 2019), path sampling (You et al., 2020; Guo et al., 2020; Xie et al., 2018), and channel pruning (Mei et al., 2020; Yu & Huang, 2019). However, recent analysis (Sciuto et al., 2020; Wang et al., 2021) shows that the magnitude of importance parameters in the weight-sharing NAS framework does not reflect the true ranking of the final architectures. Without weight-sharing, hyperparameter optimization methods (Tan & Le, 2019; Radosavovic et al., 2020; Baker et al., 2017; Wen et al., 2020; Lu et al., 2019; Luo et al., 2020; Chau et al., 2020; Yan et al., 2020) has shown its effectiveness by learning the relationship between network hyperparameters and their performance. For example, RegNet (Radosavovic et al., 2020) explains the widths and depths of good networks by a quantized linear function. Predictor-based methods (Wen et al., 2020; Luo et al., 2020; Chau et al., 2020; Yan et al., 2020; Luo et al., 2018) learn predictors, such as Gaussian process (Dai et al., 2019) and graph convolution networks (Wen et al., 2020), to predict the performance of all candidate models in the search space. A subset of models with high predicted accuracies is then trained for the final selection.

Our Network Coding Propagation (NCP) belongs to the predictor-based method, but is different from existing methods in: (1) NCP searches in a large search space in seconds without evaluating all candidate models by back-propagating the neural predictor and traversing the search space along the gradient direction. (2) Benefiting from the gradient back-propagation in network coding space, NCP is naturally applicable to various objectives across different tasks, such as multi-task structure

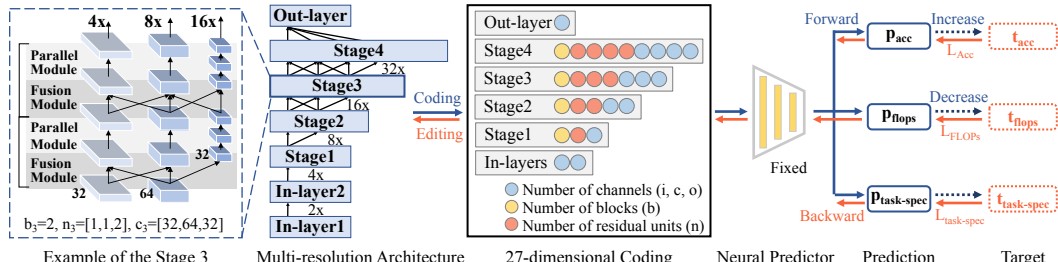

Figure 2: Overview of our Network Coding Propagation (NCP) framework. Each architecture in our search space follows a multi-resolution paradigm, where each network contains four stages, and each stage is composed of modularized blocks (a parallel module and a fusion module). The example on the left shows the 3rd stage consists of $b_3 = 2$ modularized blocks with three branches, where each branch contains a number of $n_3^i$ residual units with a channel number of $c_3^i$, $i \in \{1, 2, 3\}$. We first learn a neural predictor between an architecture coding and its evaluation metrics such as accuracy and FLOPs. After that, we set optimization objectives and back-propagate the predictor to edit the network architecture along the gradient directions.

optimization, architecture transferring, and accuracy-efficiency trade-offs. (3) Different from Luo et al. (2018); Baker et al. (2017) jointly train an encoder, a performance predictor, and a decoder to minimize the combination of performance prediction loss and structure reconstruction loss, we learn and inverse the neural predictor directly in our coding space to make the gradient updating and network editing explicit and transparent.

## 3  METHODOLOGY

NCP aims to customize specific network hyperparameters for different optimization objectives, such as single- and multi-task learning and accuracy-efficiency trade-offs, in an architecture coding space. An overview of NCP is shown in Fig. 2. In this section, we first discuss learning efficient models on a single task with two strategies and then show that it can be easily extended to multi-task scenarios. Lastly, we demonstrate our multi-resolution coding space and NAS-Bench-MR.

### 3.1  NETWORK CODING PROPAGATION

For each task, we learn a neural predictor $\mathcal{F}(\cdot)$ that projects an architecture code $e$ to its ground truth evaluation metrics, such as accuracy $g_{\text{acc}}$ and FLOPs $g_{\text{flops}}$. Given an initialized and normalized coding $e$, the predicted metrics and the loss to learn the neural predictor are represented by:

$$p_{\text{acc}}, p_{\text{flops}}, p_{\text{task-spec}} = \mathcal{F}_W(e), \tag{1}$$

$$L_{\text{predictor}} = \text{L2}(p_{\text{acc}}, g_{\text{acc}}) + \text{L2}(p_{\text{flops}}, g_{\text{flops}}) + ... \tag{2}$$

where $W$ is the weight of the neural predictor, which is fixed after learning. $p_{\text{acc}}, p_{\text{flops}}$, and $p_{\text{task-spec}}$ denote the predicted accuracy, FLOPs, and task-specific metrics (if any), L2 is the L2 norm. Note that although the FLOPs can be directly calculated from network codes, we learn it to enable the backward gradient upon FLOPs constraints for accuracy-efficiency trade-offs.

NCP edits the network coding $e$ by back-propagating the learned neural predictor $W$ to maximize/minimize the evaluation metrics (e.g., high accuracy and low FLOPs) along the gradient directions. According to the number of dimensions to be edited, **two strategies** are proposed to propagate the network coding: continuous propagation and winner-takes-all propagation, where the first one uses all the dimensions of the gradient to update $e$, while the latter uses only the one dimension with the largest gradient. Fig. 3 visualizes the network update process of our two strategies.

**Continuous Propagation.**   In continuous propagation, we set target metrics as an optimization goal and use gradient descending to automatically update the network coding $e$. Taking accuracy and FLOPs as an example, we set the target accuracy $t_{\text{acc}}$ and target FLOPs $t_{\text{flops}}$ for $e$ and calculate the loss between the prediction and the target by:

$$L_{\text{propagation}} = \text{L1}(p_{\text{acc}}, t_{\text{acc}}) + \lambda \text{L1}(p_{\text{flops}}, t_{\text{flops}}) \tag{3}$$

where L1 denotes the smooth L1 loss, $\lambda$ is the coefficient to balance the trade-off between accuracy and efficiency. The loss is propagated back through the fixed neural predictor $\mathcal{F}_W(\cdot)$, and the gradient on the coding $e$ is calculated as $\frac{\partial L}{\partial e}$ (use $\nabla e$ in following for simplicity). We update the coding

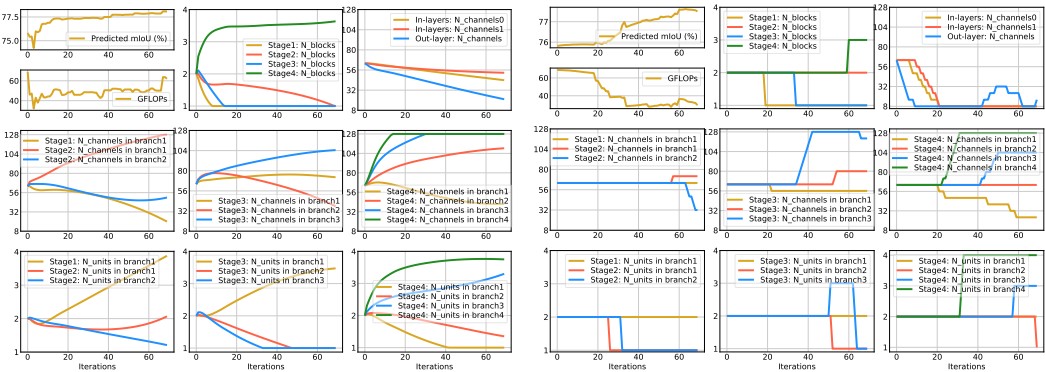

Figure 3: Visualization of the network propagation process of our two strategies (left: continuous; right: winner-takes-all) for segmentation. We group 27 dimensions into eight subfigures.

$e$ iteratively by $e \leftarrow e - \nabla e$. In this way, NCP can search for the most efficient model upon a lower-bound accuracy and the most accurate model upon upper-bound FLOPs. In practice, we set $t_{\text{acc}} = p_{\text{acc}} + 1$ and $t_{\text{flops}} = p_{\text{flops}} - 1$ to find a model with a good efficiency-accuracy trade-off.

The propagation is stopped once the prediction reaches the target or after a certain number of iterations (*e.g.*, 70). Finally, we round (the number of channels is a multiple of 8) and decode the coding $e$ to obtain the architecture optimized for the specific goal.

**Winner-Takes-All Propagation.** Editing different dimensions of the coding often have different degrees of impact on the network. For example, modifying the number of network blocks has a greater impact than the number of channels, *i.e.*, a block costs more computational resources than several channels. In light of this, we select one dimension of the coding $e$ with the best accuracy-efficiency for propagation in each iteration, termed as winner-takes-all propagation.

We create a lookup table $\mathcal{T}(\cdot)$ to obtain the corresponding FLOPs of any network coding. We then calculate the relative increase $\Delta r$ in FLOPs as each dimension of the coding grows:

$$e'_l = e_l + k$$
$$\Delta r_l = \mathcal{T}(e') - \mathcal{T}(e) \tag{4}$$

where $l$ denotes the index of the coding $e$; $k = 8$ if $e_l$ represents the number of convolutional channels, otherwise $k = 1$. We then use $\Delta r$ as a normalization term of backward gradients $\nabla e$ and select the optimal dimension $l$ of the normalized gradients $\nabla e / \Delta r$:

$$l = \begin{cases} \arg\max(\nabla e / \Delta r) & \text{if } \max(\nabla e / \Delta r) > 0 \\ \arg\min(\nabla e / \Delta r) & \text{if } \max(\nabla e / \Delta r) < 0 \end{cases} \tag{5}$$

if $\max(\nabla e / \Delta r) > 0$, modifying the value of $e_l$ may improve accuracy and decrease FLOPs simultaneously. While if $\max(\nabla e / \Delta r) < 0$, we choose the dimension with the highest improvement in accuracy under the unit FLOPs consumption. The coding $e$ is then updated as follows:

$$e_l \leftarrow \begin{cases} e_l - k & \text{if } \max(\nabla e / \Delta r) > 0 \\ e_l + k & \text{if } \max(\nabla e / \Delta r) < 0 \end{cases} \tag{6}$$

**Cross-task Learning.** Our NCP provides a natural way for cross-task learning by transforming different objectives across tasks into gradient accumulation operations directly in the architecture coding space, facilitating joint architecture search for multiple tasks and cross-task architecture transferring, as shown in Fig. 1. For example, NCP enables joint search for classification and segmentation by accumulating the gradients $\nabla e^{\text{cls}}$ and $\nabla e^{\text{seg}}$ from the two predictors:

$$\nabla e = \nabla e^{\text{cls}} + \nabla e^{\text{seg}} \tag{7}$$

Note that different weights coefficients can be used to learn multi-task architectures with task preferences. In this work, we set equal weight coefficients for these tasks.

### 3.2 NAS-BENCH-MR

**Multi-Resolution Search Space.** Inspired by HRNet (Wang et al., 2020a; Ding et al., 2021), we design a multi-branch multi-resolution search space that contains four branches and maintains both

high-resolution and low-resolution representations throughout the network. As shown in Fig. 2, after two convolutional layers decreasing the feature resolution to $1/4$ ($4\times$) of the image size, we start with this high-resolution feature branch and gradually add lower-resolution branches with feature fusions, and connect the multi-resolution branches in parallel. Multi-resolution features are resized and concatenated for the final classification/regression without any additional heads.

The design space follows the following key characteristics. (1) Parallel modules: extend each branch of the corresponding resolutions in parallel using residual units (He et al., 2016); (2) Fusion modules: repeatedly fuse the information across multiple resolutions and generate the new resolution (only between two stages). For each output resolution, the corresponding input features of its neighboring resolutions are gathered by residual units (He et al., 2016) followed by element-wise addition (upsampling is used for the low-to-high resolution feature transformation), *e.g.*, the $8\times$ output feature contains information of $4\times, 8\times$, and $16\times$ input features; (3) A modularized block is then formed by a fusion module and a parallel module.

**Coding.** We leverage the hyperparameter space of our multi-resolution network by projecting the depth and width values of all branches of each stage into a continuous-valued coding space. Formally, two $3 \times 3$ stride 2 convolutional layers with $i_1, i_2$ number of channels are used to obtain the $4\times$ high-resolution features at the beginning of the network. We then divide the network into four stages with $s = \{1, 2, 3, 4\}$ branches, respectively. Each stage contains $b_s$ modularized blocks, where the fusion module of the first block is used for transition between two stages (*e.g.*, from 2 branches to 3 branches). For the parallel module of the block, we set the number of residual units and the number of convolutional channels to $\boldsymbol{n_s} = [n_s^1, \ldots, n_s^s]$ and $\boldsymbol{c_s} = [c_s^1, \ldots, c_s^s]$ respectively, where $s$ denotes the $s$-th stage containing $s$ branches. Since the fusion module always exists between two parallel modules, once the hyperparameters of the parallel modules are determined, the corresponding number of channels of fusion modules will be set automatically to connect the two parallel modules. The $s$-th stage comprising a plurality of same modular blocks is represented by:

$$b_s, n_s^1, \ldots, n_s^s, c_s^1, \ldots, c_s^s \tag{8}$$

where the four stages are represented by 3, 5, 7, and 9-dimensional codes, respectively. By the end of the 4-th stage of the network, we use a $1 \times 1$ stride 1 convolutional layer with a channel number of $o_1$ to fuse the four resized feature representations with different resolutions. In general, the entire architecture can be represented by a 27-dimensional continuous-valued code:

$$\boldsymbol{e} = i_1, i_2, b_1, \boldsymbol{n_1}, \boldsymbol{c_1}, \ldots, b_4, \boldsymbol{n_4}, \boldsymbol{c_4}, o_1 \tag{9}$$

We then build our coding space by sampling and assigning values to the codes $\boldsymbol{e}$. For each structure, we randomly choose $b, n \in 1, 2, 3, 4$, and $i, c, o \in \{8, 16, 24, \ldots, 128\}$, resulting in a 27-dimensional coding. In this way, a fine-grained multi-resolution space that enables searching structures for various tasks with different preferences on the granularity of features is constructed.

**Dataset.** We build our NAS-Bench-MR on four carefully selected visual recognition tasks that require different granularity of features: image classification on ImageNet (Deng et al., 2009), semantic segmentation on Cityscapes (Cordts et al., 2016), 3D object detection on KITTI (Geiger et al., 2012), and video recognition on HMDB51 (Kuehne et al., 2011).

For **classification**, considering the diversity of the number of classes and data scales in practical applications, we construct three subsets: ImageNet-50-1000 (Cls-A), ImageNet-50-100 (Cls-B), and ImageNet-10-1000 (Cls-C), where the first number indicates the number of classes and the second one denotes the number of images per class. For **segmentation**, we follow HRNet (Wang et al., 2020a) by upsampling all branches to the high-resolution one. For **3D detection**, we follow PointPillars (Lang et al., 2019) that first converts 3D point points into bird's-eye view (BEV) featuremaps and then unitizes a 2D network. For **video recognition**, we replace the last 2D convolutional layer before the final classification layer of our network with a 3D convolutional layer.

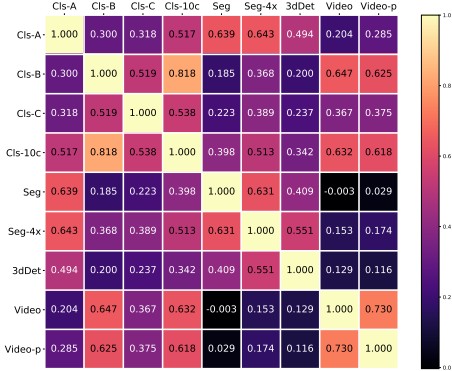

Figure 4: Spearman's rank correlation of 9 subtasks in NAS-Bench-MR.

Table 2: Performance comparison (%) on the Cityscapes validation set. All hyperparameter optimization methods are searched using our NAS-Bench-MR. All models are trained from scratch. FLOPs is measured using an input size of $512 \times 1024$. $\lambda = 0.7, 0.3, 0.1$ for 'S', 'M', and 'L'.

| Model | Type | Params | FLOPs | mIoU | mAcc | aAcc |
|---|---|---|---|---|---|---|
| ResNet34-PSP (Zhao et al., 2017) | manually-designed | 23.05M | 193.12G | 76.17 | 82.66 | 95.99 |
| ResNet50-PSP (Zhao et al., 2017) | manually-designed | 48.98M | 356.91G | 76.49 | 83.18 | 95.98 |
| HRNet-W18 (Wang et al., 2020a) | manually-designed | 9.64M | 37.01G | 77.73 | 85.61 | 96.20 |
| HRNet-W32 (Wang et al., 2020a) | manually-designed | 29.55M | 90.55G | 79.28 | 86.48 | 96.31 |
| SqueezeNAS (Shaw et al., 2019a) | NAS in SqueezeNAS space | 3.00M | 32.73G | 75.19 | – | – |
| Auto-DeepLab (Liu et al., 2019a) | NAS in DeepLab space | 10.15M | 289.78G | 79.74 | – | – |
| NetAdapt (Yang et al., 2018) | hyperparameter optimization | 10.83M | 42.49G | 78.02 | 85.68 | 96.24 |
| Random Search (Bergstra & Bengio, 2012) | hyperparameter optimization | 16.27M | 130.50G | 77.66 | 85.35 | 96.13 |
| Neural Predictor (Wen et al., 2020) | hyperparameter optimization | 33.49M | 182.60G | 78.89 | 86.46 | 96.21 |
| **NCP-Net-M-winner takes all** | hyperparameter optimization | 7.18M | 32.88G | 77.91 | 85.69 | 96.23 |
| **NCP-Net-M-continuous** | hyperparameter optimization | 8.39M | 36.25G | 78.36 | 86.24 | 96.29 |
| **NCP-Net-S-continuous** | hyperparameter optimization | 6.04M | 20.07G | 76.98 | 85.14 | 96.07 |
| **NCP-Net-L-continuous** | hyperparameter optimization | 37.70M | 179.98G | **80.05** | **87.12** | **96.49** |

We also explore several proxy training settings, such as 10-epoch training used in RegNet (Radosavovic et al., 2020) (Cls-10c), reduced input resolution by four times (Seg-4x), and pretraining instead of training from scratch (Video-p). In summary, we randomly sample 2,500 structures from our search space, train and evaluate these same architectures for those above nine different tasks/settings, resulting in 22,500 trained models. We believe these well-designed and fully-trained models serve as an important contribution of this work to facilitate future NAS research.

## 4 EXPERIMENTS

### 4.1 IMPLEMENTATION DETAILS

Taking a three-layer fully connected network as the neural predictor, we apply NCP to NAS-Bench-MR and obtain NCP-Net. To train the neural predictor, 2000 and 500 structures in the benchmark are used as the training and validation sets for each task. Unless specified, we use continuous propagation with an initial code of $\{b, n = 2; c, i, o = 64\}$ and $\lambda = 0.5$ for 70 iterations in all experiments. The optimization goal is set to higher performance and lower FLOPs ($t_{acc} = p_{acc} + 1$, $t_{flops} = p_{flops} - 1$). More experiments and details can be found in **Appendix**.

### 4.2 BASELINE EVALUATIONS

We make comparisons among different search strategies on our multi-resolution space and segmentation benchmark in Tab. 2. For Random Search (Bergstra & Bengio, 2012) and Neural Predictor (Wen et al., 2020), we sample 100 and 10,000 network codes respectively and report the highest result in top-10 models. For NetAdapt (Yang et al., 2018), we traverse each dimension of the initial codes (increasing or decreasing a unit value) and adapt the codes by greedy strategy.

**Effectiveness of NCP.** (1) NCP outperforms manually-designed networks, *e.g.*, ResNet (He et al., 2016) and HRNet (Wang et al., 2020a), hyperparameter optimization methods (Bergstra & Bengio, 2012; Wen et al., 2020; Yang et al., 2018), and even well-designed weight-sharing NAS (Liu et al., 2019a; Shaw et al., 2019a). (2) Compared to other hyperparameter optimization methods, NCP is not sensitive to the randomness and noise in model selection by adopting an approximate gradient direction. (3) NCP is much more efficient as it evaluates all dimensions of the codes with only one back-propagation without top-K ranking. (4) NCP is able to search for optimal architectures under different computing budgets by simply setting the coefficient $\lambda$.

**Effectiveness of our search space.** (1) Random Search (Bergstra & Bengio, 2012) works well in our coding space, showing the effectiveness of our multi-resolution space. (2) Models in our search space consist of only $3 \times 3$ convolution in the basic residual unit (He et al., 2016) without complex operators, making it applicable to various computing platforms, such as CPU, GPU, and FPGA.

**Search strategies of NCP.** Intuitively, the winner-take-all strategy is more like a greedy algorithm. It selects one coding dimension with the best accuracy-efficiency trade-off for propagation, resulting

Table 3: Performance (%) on five datasets of the models searched on different optimization objectives (including single- and multi-task optimization) using our NCP, *e.g.* "Cls + Seg" denotes the model is searched using "Cls-A" and "Seg" benchmarks, "Four Tasks" denotes the model is propagated by using the predictors trained on the all four benchmarks (Cls-A, Seg, 3dDet, Video). Note that in addition to cross-task evaluation, we also show the generalizability of NAS by applying the searched model to a new dataset, ADE20K, which is not used to train neural predictors. For clearer comparisons, the FLOPs of all networks is measured using input size $128 \times 128$ under the segmentation task. The top-2 results are highlighted in **bold**.

| Method | Params | FLOPs | Classification ImageNet-50-1000 | | Segmentation Cityscapes | | | Video HMDB51 | | 3D Object Detection KITTI | | | | Segmentation ADE20K | | |
|---|---|---|---|---|---|---|---|---|---|---|---|---|---|---|---|---|
| | | | top1 | top5 | mIoU | mAcc | aAcc | top1 | top5 | car-3D | car-BEV | ped-3D | ped-BEV | mIoU | mAcc | aAcc |
| Cls | 5.35M | 2.05G | **85.56** | **95.36** | 75.65 | 84.11 | 95.23 | 25.53 | 59.69 | 73.60 | 83.92 | 35.20 | 41.39 | 32.74 | 42.77 | 76.99 |
| Seg | 7.91M | 0.90G | 82.52 | 94.00 | **77.15** | **84.73** | **96.01** | 14.68 | 44.92 | 69.99 | 84.12 | 21.83 | 31.21 | 33.32 | 42.97 | 77.41 |
| Video | 2.56M | 0.74G | 81.88 | 93.84 | 71.09 | 79.91 | 95.14 | **28.47** | **62.18** | 68.60 | 81.68 | 19.02 | 27.82 | 25.84 | 34.07 | 73.88 |
| 3dDet | 2.89M | 1.21G | 82.60 | 94.64 | 69.43 | 78.48 | 95.05 | 19.84 | 53.82 | 75.44 | **87.37** | 40.42 | **48.69** | 23.28 | 30.64 | 72.29 |
| Cls + Seg | 8.29M | 1.70G | **86.05** | **95.35** | **77.16** | **84.95** | **96.18** | 22.95 | 57.74 | 74.39 | 84.72 | 27.18 | 35.70 | **35.58** | **46.02** | **78.19** |
| Cls + Video | 4.32M | 1.60G | 85.64 | 95.28 | 72.98 | 81.51 | 95.54 | 27.83 | 60.84 | 71.05 | 84.56 | 23.84 | 30.27 | 30.43 | 41.29 | 75.78 |
| Cls + 3dDet | 4.46M | 1.81G | 85.76 | 95.27 | 72.18 | 80.81 | 95.42 | 23.17 | 58.40 | **75.47** | **87.63** | **40.61** | **48.80** | 31.98 | 42.00 | 76.67 |
| Seg + Video | 4.46M | 1.16G | 83.84 | 94.20 | 75.64 | 84.16 | 95.88 | **28.46** | 60.77 | 71.24 | 83.47 | 34.44 | 42.00 | 34.23 | 44.45 | 77.28 |
| Seg + 3dDet | 4.30M | 0.91G | 83.08 | 94.57 | 75.60 | 83.74 | 95.85 | 21.01 | 56.46 | **75.54** | 86.67 | **41.45** | 46.32 | **34.65** | **46.38** | **77.71** |
| Video + 3dDet | 2.37M | 0.85G | 83.27 | 94.66 | 70.32 | 79.33 | 95.12 | 28.20 | **61.15** | 74.83 | 83.52 | 37.82 | 44.17 | 26.42 | 36.34 | 73.69 |
| Four Tasks | 4.40M | 1.44G | 84.36 | 95.04 | 75.49 | 83.86 | 95.83 | 25.09 | 58.36 | 71.90 | 84.29 | 27.27 | 34.34 | 34.53 | 45.36 | 77.45 |

in efficient models. The continuous strategy updates all dimensions of the coding purely based on gradients. Thus, it reaches better accuracy, though the generated model is not the most efficient. The experiment verifies the above intuition. Unless specified, we use the continuous strategy for all experiments. Fig. 3 shows the code editing process of two strategies.

## 4.3 INTER-TASK ARCHITECTURE SEARCH

We conduct multi-task architecture learning experiments on four basic benchmarks, including Cls-A, Seg, Video, and 3dDet. We use NCP to search models on single (*e.g.*, Cls) or multiple tasks (*e.g.*, Cls + Seg) and evaluate their performance on five datasets, including a new dataset ADE20K (Zhou et al., 2017) which is not included in NAS-Bench-MR. Quantitative comparisons are shown in Tab. 3.

**Optimization for various objectives.** (1) NCP can customize good architectures for every single task, *e.g.*, the model searched on Cls achieves a top-1 accuracy of 85.56%, better than the models searched on the other three tasks. (2) NCP can learn architectures that achieve good performance on multiple tasks by inverting their predictors simultaneously, *e.g.*, the model searched on Cls + Seg achieves a top1 accuracy of 86.05% and 77.16% mIoU. The joint optimization of all four tasks achieved moderate results on each task. (3) NCP achieves good accuracy-efficiency trade-offs and tunes a proper model size for each task automatically by using the FLOPs constraint.

**Relationship among tasks.** Two tasks may help each other if they are highly related, and vice versa. For example, jointly optimizing Seg + Cls (correlation is 0.639) improves the performance on both tasks with 17% fewer FLOPs than a single Cls model, while jointly optimizing Seg + Video (correlation is -0.003) hinders the performance of both tasks. When optimizing Seg + 3dDet (correlation is 0.409) simultaneously, the resulted network has better results in 3dDet but worse in Seg, which means accurate semantic information is useful for 3D detection, while the object-level localization may harm the performance of per-pixel classification. The above observations can be verified by Spearman's rank correlation coefficients in Fig. 4. This way, we can decide which tasks are more suitable for joint optimization, such as CLs + Video, improving both tasks.

**Generalizability.** The searched models work well on the new dataset ADE20K (Zhou et al., 2017). An interesting observation is that the searched models under multi-task objectives, *e.g.*, Seg + Cls, even conflicting tasks, *e.g.*, Seg + Video, show better performance on ADE20K than those searching on a single task, demonstrating the generalizability of NCP. We also adapt our segmentation model (NCP-Net-L-continuous) in Tab. 2 to the COCO instance segmentation dataset. The model with only 360 GFLOPs (measured using 800x1280 resolution, Mask R-CNN head) achieves 47.2% bounding box AP and 42.3% mask AP on the detection and instance segmentation tasks, respectively. With fewer computational costs, our results outperform HRNet-W48 (46.1% for bbox AP and 41.0% for mask AP) by a large margin.

Table 4: Our architecture transfer results between four different tasks. We first find an optimal architecture coding for each task and then use it as the initial coding to search other three tasks. "-F" and "-T" denote the architecture finetuning and transferring results, respectively.

| Method | FLOPs-F | Classification ImageNet-50-1000 | | | Segmentation Cityscapes | | | Video Recognition HMDB51 | | | 3D Object Detection KITTI | | |
|---|---|---|---|---|---|---|---|---|---|---|---|---|---|
| | | top1-F | top1-T | FLOPs-T | mIoU-F | mIoU-T | FLOPs-T | top1-F | top1-T | FLOPs-T | car-3D-F | car-3D-T | FLOPs-T |
| Classification | 2.05G | **85.56** | – | – | 75.42 | **78.27** | 1.10G | 25.53 | **28.65** | 0.76G | 73.60 | 75.59 | 1.30G |
| Segmentation | 0.90G | 82.52 | 85.84 | 1.54G | **77.15** | – | – | 14.68 | 28.44 | 0.75G | 69.99 | **76.17** | 1.21G |
| Video Recognition | 0.74G | 81.88 | **86.03** | 1.86G | 71.09 | 77.56 | 0.95G | **28.47** | – | – | 68.60 | 75.78 | 1.17G |
| 3D Object Detection | 1.21G | 82.60 | 85.63 | 1.61G | 69.43 | 77.25 | 1.04G | 19.84 | 28.12 | 0.78G | **75.44** | – | – |

## 4.4 Cross-task Architecture Transferring

We experiment on architecture transferring across four different tasks. We first find optimal network codings for each task and then use it as the initial coding to search each of the other three tasks.

Comparison results can be found in Tab. 4. We see that: (1) Compared to searching a network from scratch for one task (*e.g.*, 77.15% on segmentation) or finetuning architectures trained on other tasks (*e.g.*, 75.42% by finetuning a classification model), our transferred architecture are always better (*e.g.*, 78.27% by transferring a classification architecture to segmentation).

(2) The architecture searched on classification achieves better transferable ability on segmentation and video recognition, while the segmentation model is more suitable for transferring to 3D detection. It can be validated by Spearman's rank correlation in Fig. 4.

(3) NCP is not sensitive to the initialized architecture coding. Models can be searched with different initial codings and achieve good performance.

## 4.5 Intra-task Architecture Search

To show NCP 's ability to customize architectures according to different data scales and the number of classes, we evaluate the searched architecture on the three subsets of the ImageNet dataset. From Tab. 5, we observe that: (1) NCP can find optimal architectures for each subset, which is essential for practical applications with different magnitudes of data. (2) We perform a joint architecture search for all three subsets. The searched architecture (NCP-Net-ABC) shows surprisingly great results on all three subsets with only 0.77G extra FLOPs and fewer parameters, demonstrating intra-task generalizability of NCP.

Table 5: Single-crop top-1 error rates (%) on the ImageNet validation set. 'ImNet-A', 'ImNet-B', 'ImNet-C' denote the predictor is trained on benchmarks of ImageNet-50-1000, ImageNet-50-100, ImageNet-10-1000, respectively. FLOPs is measured using input size $224 \times 224$. $\lambda = 0.7$. Top-2 results are highlighted in **bold**.

| Model | Params | FLOPs | ImNet-A | ImNet-B | ImNet-C |
|---|---|---|---|---|---|
| ResNet50 | 23.61M | 4.12G | 83.76 | 50.16 | 86.00 |
| ResNet101 | 42.60M | 7.85G | 84.32 | 51.92 | 86.00 |
| HRNet-W32 | 39.29M | 8.99G | 84.00 | 52.00 | 83.80 |
| HRNet-W48 | 75.52M | 17.36G | 84.52 | 52.88 | 86.60 |
| **NCP-Net-A** | 4.39M | 5.95G | **85.80** | 56.04 | 86.80 |
| **NCP-Net-B** | 3.33M | 3.55G | 82.76 | **56.40** | 85.60 |
| **NCP-Net-C** | 4.06M | 5.51G | 84.60 | 55.60 | **87.80** |
| **NCP-Net-ABC** | 4.28M | 6.72G | **85.12** | **58.12** | **88.20** |

## 5 Conclusion

This work provides an initial study of learning task-transferable architectures in the network coding space. We propose an efficient NAS method, namely Network Coding Propagation (NCP), which optimizes the network code to achieve the target constraints with back-propagation on neural predictors. In NCP, the multi-task learning objective is transformed to gradient accumulation across multiple predictors, making NCP naturally applicable to various objectives, such as multi-task structure optimization, architecture transferring across tasks, and accuracy-efficiency trade-offs. To facilitate the research in designing versatile network architectures, we also build a comprehensive NAS benchmark (NAS-Bench-MR) upon a multi-resolution network space on many challenging datasets, enabling NAS methods to spot good architectures across multiple tasks, other than classification as the main focus of prior works. We hope this work and models can advance future NAS research.

## ACKNOWLEDGEMENTS

We sincerely appreciate all reviewers' efforts and constructive suggestions in improving our paper. Ping Luo was supported by the General Research Fund of HK No.27208720 and HKU-TCL Joint Research Center for Artificial Intelligence. Zhiwu Lu was supported by National Natural Science Foundation of China (61976220).

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

Table 6: Detailed statistics of our NAS-Bench-MR. NAS-Bench-MR contains 4 datasets and 9 settings. Considering the diversity and complexity of real-world applications (*e.g.* different scales/input sizes of training data), we use a variety of challenging settings (*e.g.*, full resolution and training epochs) to ensure that the model is fully trained. For classification, we train models with different numbers of classes, numbers of training samples, and training epochs. For semantic segmentation, we train models under different input sizes. We also evaluated video action recognition models under two settings: trained from scratch and pretrained with ImageNet-50-1000. † denotes each sample contains multiple classes. ‡ denotes there are three basic classes (car, pedestrian, cyclist) in KITTI, while for each object we also regress its 3D location (XYZ), dimensions (WHL), and orientation ($\alpha$). "N" – training from scratch. "Y" – training with the ImageNet-50-1000 pretrained model. We also give the mean, std, and the validation L1 loss (%) of the neural predictor under the main evaluation metric of each setting.

| | Cls-50-1000 | Cls-50-100 | Cls-10-1000 | Cls-10c | Seg | Seg-4x | 3dDet | Video | Video-p |
|---|---|---|---|---|---|---|---|---|---|
| Dataset | ImNet-50-1000 | ImNet-50-100 | ImNet-10-1000 | ImNet-50-1000 | Cityscapes | resized Cityscapes | KITTI | HMDB51 | HMDB51 |
| Input Size | $224 \times 224$ | $224 \times 224$ | $224 \times 224$ | $224 \times 224$ | $512 \times 1024$ | $128 \times 256$ | $432 \times 496$ | $112 \times 112$ | $112 \times 112$ |
| Epochs | 100 | 100 | 100 | 10 | 530 | 530 | 80 | 100 | 100 |
| Number of classes | 50 | 50 | 10 | 50 | 19 | 19 | 3‡ | 51 | 51 |
| Number of training samples per class | 1000 | 1000 | 1000 | 1000 | 2975† | 2975† | 3712† | $\approx 70$ | $\approx 70$ |
| Pretrained? | N | N | N | N | N | N | N | N | Y |
| Main metric | top1 Acc | top1 Acc | top1 Acc | top1 Acc | mIoU | mIoU | car-3D AP | top1 Acc | top1 Acc |
| Mean | 82.21 | 48.59 | 85.70 | 57.07 | 72.58 | 62.21 | 76.36 | 19.56 | 30.88 |
| Standard Deviation | 2.64 | 3.69 | 1.51 | 4.51 | 5.33 | 3.37 | 4.70 | 3.69 | 4.65 |
| Prediction Loss | 0.56 | 1.07 | 0.83 | 1.05 | 0.75 | 0.88 | 0.85 | 1.07 | 1.26 |

# A DETAILS OF NAS-BENCH-MR

In this section, we provide details of the datasets and settings used to build our NAS-Bench-MR. We follow the common practice and realistic settings for training and evaluation in each task, making the scientific insights generated by our benchmark easier to generalize to real-world scenarios.

## A.1 DATASETS AND SETTINGS

We randomly sample 2,500 structures from our network coding space, and train and evaluate these same architectures for each task and setting. Each architecture is represented by a 27-dimensional continuous-valued code. Tab. 7 shows the representation of each dimension of the code. For 9 different datasets and proxy tasks, we train a total of 22,500 models. See below for details.

**ImageNet for Image Classification.** The ILSVRC 2012 classification dataset (Deng et al., 2009) consists of 1,000 classes, with a number of 1.2 million training images and 50,000 validation images.

Considering the diversity of the number of classes and data scales in practical applications, and saving training time, we construct three subsets: ImageNet-50-1000, ImageNet-50-100, and ImageNet-10-1000, where the first number indicates the number of classes and the second one denotes the number of images per class.

In this work, we adopt an SGD optimizer with momentum 0.9 and weight decay 1e-4. The input size is $224 \times 224$. The initial learning rate is set to 0.1 with a total batch size of 160 on 2 Tesla V100 GPUs for 100 epochs, and decays by cosine annealing with a minimum learning rate of 0. We adopt the basic data augmentation scheme to train the classification models, i.e., random resizing and cropping, and random horizontal flipping (flip ratio 0.5), and use single-crop for evaluation.

We report the top-1 and top-5 accuracies as the evaluation metric on all three benchmark datasets. We print the training log (top-1, top-5, loss, lr) every 20 iterations and evaluate the model every 5 epochs on the ImageNet validation set. We will release four checkpoints of 25, 50, 75, 100 epochs with the optimizer data and all the training logs for each model.

**Cityscapes for Semantic Segmentation.** The Cityscapes dataset (Cordts et al., 2016) contains high-quality pixel-level annotations of 5000 images with size $1024 \times 2048$ (2975, 500, and 1525 for the training, validation, and test sets respectively) and about 20000 coarsely annotated training images. Following the evaluation protocol (Cordts et al., 2016), 19 semantic labels are used for evaluation without considering the void label.

Table 7: Representations of our 27-dimensional coding.

| Component | Codes | | |
| | $N_{blocks}$ | $N_{residual\ units}$ | $N_{channels}$ |
| --- | --- | --- | --- |
| Input layers | | | $i_1, i_2$ |
| Stage 1 | $b_1$ | $n_1^1$ | $c_1^1$ |
| Stage 2 | $b_2$ | $n_2^1, n_2^2$ | $c_2^1, c_2^2$ |
| Stage 3 | $b_3$ | $n_3^1, n_3^2, n_3^3$ | $c_3^1, c_3^2, c_3^3$ |
| Stage 4 | $b_4$ | $n_4^1, n_4^2, n_4^3, n_4^4$ | $c_4^1, c_4^2, c_4^3, c_4^4$ |
| Output layer | | | $o_1$ |

To study the effect of different image resolutions in segmentation, we conduct two benchmarks with the input size of $512 \times 1024$ and $128 \times 256$, respectively. For the small-resolution setting, we pre-resize the Cityscapes dataset to $256 \times 512$ before the data augmentation.

In this work, we use an SGD optimizer with momentum 0.9 and weight decay 4e-5. The initial learning rate is set to 0.1 with a total batch size of 64 on 8 Tesla V100 GPUs for 25000 iterations (about 537 epochs). Follows the common practice in (Zhao et al., 2017; Wang et al., 2020a; Ding et al., 2020b), the learning rate and momentum follow the "poly" scheduler with power 0.9 and a minimum learning rate of 1e-4. We use basic data augmentation, *i.e.*, random resizing and cropping, random horizontal flipping, and photometric distortion for training and single-crop testing with the test size of $1024 \times 2048$ and $256 \times 512$ respectively for two settings.

We report the mean Intersection over Union (mIoU), mean (macro-averaged) Accuracy (mAcc), and overall (micro-averaged) Accuracy (aAcc) as the evaluation metrics. We print the training log (mAcc, loss, lr) every 50 iterations and evaluate the model every 5000 iterations on the Cityscapes validation set. We will release five checkpoints of 5000, 10000, 15000, 20000, 25000 iterations with the optimizer data and all the training logs for each model.

**KITTI for 3D Object Detection.** The KITTI 3D object detection dataset (Geiger et al., 2012) is widely used for monocular and LiDAR-based 3D detection. It consists of 7,481 training images and 7,518 test images as well as the corresponding point clouds and the calibration parameters, comprising a total of 80,256 2D-3D labeled objects with three object classes: Car, Pedestrian, and Cyclist. Each 3D ground truth box is assigned to one out of three difficulty classes (easy, moderate, hard) according to the occlusion and truncation levels of objects.

In this work, we follow the train-val split (Chen et al., 2015; Ding et al., 2020a), which contains 3,712 training and 3,769 validation images. The overall framework is based on Pointpillars (Lang et al., 2019). The input point points are projected into bird's-eye view (BEV) feature maps by a voxel feature encoder (VFE). The projected BEV feature maps ($496 \times 432$) are then used as the input of our 2D network for 3D/BEV detection.

Following (Lang et al., 2019), we set, pillar resolution: 0.16m, max number of pillars: 12000, and max number of points per pillar: 100. We apply the same data augmentation, *i.e.*, random mirroring and flipping, global rotation and scaling, and global translation for 3D point clouds as in Pointpillar (Lang et al., 2019). We use the one-cycle scheduler with an initial learning rate of 2e-3, a minimum learning rate of 2e-4, and batch size 16 on 8 Tesla V100 GPUs for 80 epochs. We use an AdamW optimizer with momentum 0.9 and weight decay 1e-2. At inference time, axis-aligned non-maximum suppression (NMS) with an overlap threshold of 0.5 IoU is used for final selection.

We report standard average precision (AP) on each class as the evaluation metric. We will release five checkpoints of 76, 77, 78, 79, 80 epochs with the optimizer data and detailed evaluation results (Car/Pedestrian/Cyclist easy/moderate/hard 3D/BEV/Orientation detection AP) of each checkpoint for each model. Due to the unstable training of the KITTI dataset, we provide the last five checkpoints for researchers to tune hyperparameters from different criteria, such as the single model best performance, the average best performance, and the last epoch best performance.

**HMDB51 for Video recognition.** We train video recognition models on the HMDB51 dataset (Kuehne et al., 2011), consisting of 6,766 videos with 51 categories. We use the first training and validation split composing of 3570 and 1530 videos for evaluation, respectively. The video containing less than 64 frames is filtered, resulting in 2649 samples for training.

Considering the difficulty of learning temporal information, in addition to training from scratch, we also conduct training using models of ImageNet-50-1000 as pretraining. To model the temporal

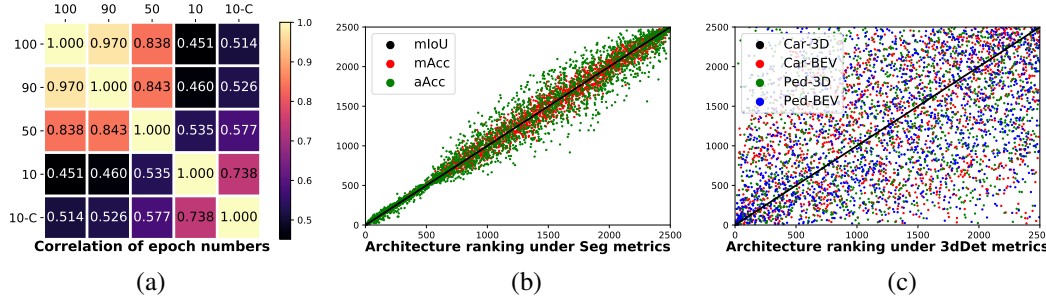

Figure 5: (a) Spearman's rank correlation of the different number of epochs (checkpoints of 10, 50, 90, 100 epochs during training on ImageNet-50-1000). '10-C' denotes using the convergent learning rate for 10 epochs, *i.e.*, the proxy setting used in RegNet (Radosavovic et al., 2020). (b) Architecture rankings under the evaluation metrics of semantic segmentation (mIoU, mAcc, aAcc). (c) Architecture rankings under the evaluation metrics of 3D object detection (car/pedestrian 3D/bird's-eye view detection AP).

information, we replace the last 2D convolutional layer before the final classification layer of HRNet and its counterparts with a 3D convolutional layer.

In this work, the input size is set to $112 \times 112$. We adopt an Adam optimizer with momentum 0.9 and weight decay 1e-5. The initial learning rate is set to 0.01 with a total batch size of 80 on 4 Tesla V100 GPUs for 100 epochs, and decays by cosine annealing with a minimum learning rate of 0. We adopt random resizing and cropping, random brightness 0.5, random contrast 0.5, random saturation 0.5, random hue 0.25, and random grayscale 0.3 as data augmentation.

We report the top-1 and top-5 accuracies as the evaluation metric on both two settings. We print the training log (top-1, top-5, loss, lr) every 10 iterations and evaluate the model every 10 epochs on the validation set. We will release the last checkpoint and all the training logs for each model.

## A.2 ANALYSIS OF NAS-BENCH-MR

We summarize detailed statistics of our NAS-Bench-MR in Tab. 6. We also give the mean, std, and the validation L1 loss (%) of our neural predictors under the main evaluation metric of each setting.

From Spearman's rank correlation of nine benchmarks in our main paper, we observe that:

(1) The four main tasks have different preferences for the architecture, and their correlation coefficients are between -0.003 (segmentation and video recognition) and 0.639 (segmentation and classification). This may be because segmentation can be seen as per-pixel image classification.

(2) Different settings in the same task also have different preferences for the architecture, and their correlation coefficients are between 0.3 (Cls-50-1000 and Cls-50-100) and 0.818 (Cls-50-100 and Cls-10c). From Fig. 5 (a) we also see that different training epochs result in a significant performance change. In this way, the proxy 10-epoch training setting (Radosavovic et al., 2020) may not generalize well to real-world scenarios.

(3) The correlation coefficient matrix is related to the model performance of multi-task joint optimization using NP. The higher the correlation coefficient of the two tasks, the greater the gain of joint optimization. If the correlation coefficient of the two tasks is too low, joint optimization may reduce the performance on both tasks.

(4) We show the architecture ranking under different metrics for different tasks in Fig. 5 (b) and (c). We noticed that the three metrics (*i.e.*, mIoU, mAcc, aAcc) in segmentation are positively correlated, which means only one (*e.g.*, mIoU) needs to be optimized to obtain a good model under all three metrics. However, in the 3D object detection task, different metrics are not necessarily related, which makes multi-objective optimization especially important.

Table 8: Comparative results (%) of NCP on the Cityscapes validation set. The first two architectures are searched using neural predictors that are trained on the Cityscapes dataset with an input size of $512 \times 1024$ (Seg) and the resized Cityscapes dataset with an input size of $128 \times 256$ (Seg-4x), respectively. The last model is searched by joint optimization of both the two predictors. $\lambda$ is set to 0.5 during the network codes propagation process. All models are trained from scratch. FLOPs is measured using $512 \times 1024$.

| Method | Params | FLOPs | Cityscapes $512 \times 1024$ | | | Cityscapes $128 \times 512$ | | | ADE20K $512 \times 512$ | | |
|---|---|---|---|---|---|---|---|---|---|---|---|
| | | | mIoU | mACC | aACC | mIoU | mACC | aACC | mIoU | mACC | aACC |
| ResNet18-PSP (Zhao et al., 2017) | 12.94M | 108.53G | 73.21 | 79.86 | 95.51 | 62.66 | 70.17 | 93.25 | 33.45 | 40.95 | 77.78 |
| ResNet34-PSP (Zhao et al., 2017) | 23.05M | 193.12G | 76.17 | 82.66 | 95.99 | 64.17 | 72.37 | 93.48 | 32.78 | 41.23 | 77.52 |
| HRNet-W18s (Wang et al., 2020a) | 5.48M | 22.45G | 76.21 | 84.43 | 95.86 | 64.49 | 73.58 | 93.71 | 34.39 | 44.65 | 77.60 |
| HRNet-W18 (Wang et al., 2020a) | 9.64M | 37.01G | **77.73** | **85.61** | 96.20 | 65.19 | 74.85 | 93.66 | 34.84 | 45.27 | 77.80 |
| **NCP-Net-$512 \times 1024$** (Fig. 19) | 7.91M | 29.34G | 77.15 | 84.73 | 96.01 | 62.40 | 71.70 | 93.34 | 33.32 | 42.97 | 77.41 |
| **NCP-Net-$128 \times 512$** (Fig. 20) | 2.61M | 31.37G | 70.91 | 80.20 | 95.21 | **65.89** | **75.19** | 93.79 | 27.52 | 35.88 | 73.85 |
| **NCP-Net-Both** (Fig. 21) | 7.82M | 50.90G | 77.35 | 84.65 | **96.23** | 64.98 | 73.82 | **93.83** | **35.65** | **45.64** | **77.94** |

Table 9: Video recognition results (%) of NCP on the HMDB51 dataset. The first two models are searched using neural predictors that are trained on HMDB51 from scratch (Video) and using models of ImageNet-50-1000 as pretraining (Video-p). The last model is searched by joint optimization of both two predictors. We also show the classification accuracy of the pretrained model on the ImageNet-50-1000 dataset (the column of "Cls-Pretraining"). Note that the last 2D convolutional layer before the final classifier is replaced with a 3D convolutional layer ($3 \times 3 \times 3$) for all models to capture the temporal information. $\lambda$ is set to 0.5. FLOPs is calculated using an input size of $112 \times 112$.

| Method | Params | FLOPs | Video-Scratch | | Cls-Pretraining | | Video-Pretrained | |
|---|---|---|---|---|---|---|---|---|
| | | | top1 | top5 | top1 | top5 | top1 | top5 |
| ResNet18 (He et al., 2016) | 12.10M | 0.59G | 21.50 | 54.93 | 82.72 | 93.88 | 27.84 | 60.50 |
| ResNet34 (He et al., 2016) | 22.21M | 1.20G | 19.89 | 52.94 | 83.40 | 94.28 | 31.22 | 63.53 |
| HRNet-W18s (Wang et al., 2020a) | 14.35M | 0.86G | 24.55 | 55.87 | 83.04 | **95.16** | 30.96 | 61.21 |
| HRNet-W18 (Wang et al., 2020a) | 20.05M | 1.41G | 24.11 | 55.33 | 83.48 | 94.04 | 32.48 | 64.30 |
| **NCP-Net-Scratch** (Fig. 22) | 2.56M | 0.69G | **28.47** | 62.18 | 81.88 | 93.84 | 37.54 | 68.50 |
| **NCP-Net-Pretrained** (Fig. 23) | 1.72M | 1.00G | 27.49 | 62.01 | 82.48 | 94.80 | 39.14 | 69.66 |
| **NCP-Net-Both** (Fig. 24) | 2.39M | 1.16G | 27.14 | **62.28** | **83.92** | 94.76 | **40.21** | **70.29** |

## B  INTRA-TASK GENERALIZABILITY

In this section, we explore the effect of multiple proxy settings such as different input sizes ($512 \times 1024$ and $128 \times 256$), and pretraining strategies (pretraining from classification or training from the scratch for video recognition) in designing network architectures. We search architectures under single (*e.g.*, $512 \times 1024$) or multiple (*e.g.*, both resolutions) to validate the effectiveness of NCP for different intra-task settings.

Tab. 8 and 9 show the intra-task cross-setting generalizability of our searched NCP-Net and some manually-designed networks, such as ResNet (He et al., 2016) and HRNet (Wang et al., 2020a), on the Cityscapes dataset and the HMDB51 dataset, respectively. We see that:

(1) Generally, the network searched on a specific setting performs the best under this setting but poor under other settings, which means that NCP can optimize and customize model structures for a specific setting. For example, Tab. 8 shows NCP customizing structures for different input resolutions on Cityscapes, which is essential for practical applications in real-world scenarios.

(2) Benefiting from the high correlation between different settings of the same task, joint optimization of multiple settings within the same task (for both segmentation and video recognition) usually has a positive effect, *i.e.*, improving the performance on each setting, at the cost of the increasing model size (larger FLOPs). However, different tasks may not necessarily correlate, *e.g.* segmentation + video recognition. This can be verified by Spearman's correlation, as discussed in our main paper.

(3) In Tab. 8, we apply the searched segmentation network to the ADE20K dataset (Zhou et al., 2017) to show the generalizability of the searched architectures. The network trained with a large input resolution ($512\times1024$) has better performance than the network trained with a small resolution ($128 \times 256$).

Moreover, the jointly searched architecture using both resolutions shows better generalizability than those two with a single objective. It also demonstrated that the network searched on our NAS-Bench-MR has stronger transfer capability to real-world scenarios, compared to the previous benchmarks such as (Dong & Yang, 2020; Ying et al., 2019) that uses a small input size.

(4) For both the segmentation and the video recognition tasks, our joint searched networks outperform manually-designed networks, such as HRNet (Wang et al., 2020a) and ResNet (He et al., 2016), and achieve better generalizability to other settings and datasets, *e.g.*, ADE20K.

## C  NCP ON NAS-BENCH-201

In this section, we apply our NCP to the NAS-Bench-201 benchmark (Dong & Yang, 2020) to show its effectiveness. We first briefly introduce NAS-Bench-201 and then state the difference between our NAS-Bench-MR and NAS-Bench-201. Lastly, we detail the experiment.

### C.1  NAS-BENCH-201

NAS-Bench-201 employs a DARTS-like (Liu et al., 2019b) search space including three stacks of cells, connected by a residual block (He et al., 2016). Each cell is stacked $N = 5$ times, with the number of output channels as 16, 32, and 64 for the first, second, and third stages, respectively. The intermediate residual block is the basic residual block with a stride of 2. There are 6 searching paths in the space of NAS-Bench-201, where each path contains 5 options: (1) zeroize, (2) skip connection, (3) 1-by-1 convolution, (4) 3-by-3 convolution, and (5) 3-by-3 average pooling layer, resulting in 15,625 different models in total.

NAS-Bench-201 train these models on the ImageNet-16 (with an input size of $16\times16$) and Cifar10 (with an input size of $32 \times 32$) datasets. Since the ImageNet dataset is closer to practical applications, we use the models in ImageNet-16 as a comparison in the following section.

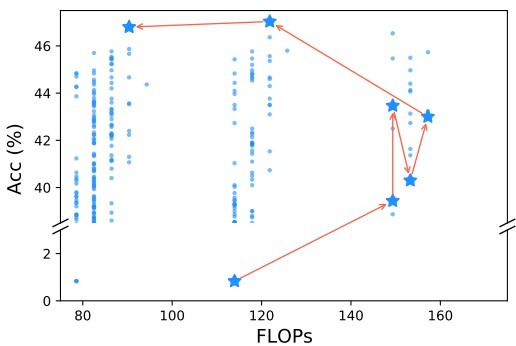

Figure 6: Visualization of the architecture propagation process on the NAS-Bench-201 benchmark (Dong & Yang, 2020) (ImageNet-16). ★ represents the propagated model in each step. NCP finds the optimal structure from a low-Acc and high-FLOPs starting point with only 6 steps by optimizing accuracy and FLOPs constraints simultaneously.

### C.2  DIFFERENCES

There are some differences between the NAS-Bench-201 benchmark and our NAS-Bench-MR.

(1) NAS-Bench-201 and NAS-Bench-MR are of different magnitudes (15,625 v.s. $10^{23}$). (2) The number of channels/blocks/resolutions is fixed in NAS-Bench-201, while it is searchable in our search space. In this way, our search space is more fine-grained and suitable for customizing to tasks with different preferences (high- and low-level features, deeper and shallower networks, wider and narrower networks, etc.). (3) The architecture in NAS-Bench-201 is represented as a one-hot encoding (choose one of five operators). While in our NAS-Bench-MR, the continuous-valued code is more appealing to the learning of the neural predictor. (4) NAS-Bench-201 is built based on a single setting while NAS-Bench-MR contains multiple settings.

We then conduct experiments and show that despite the above many differences, NCP works well on NAS-Bench-201, demonstrating the generalization ability of NCP to other benchmarks.

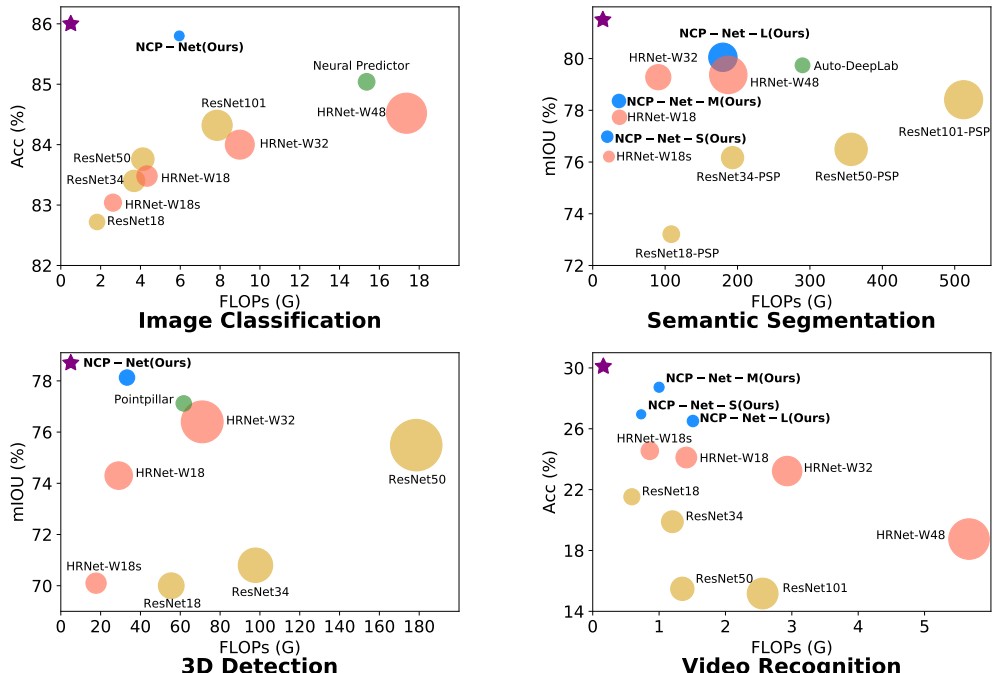

Figure 7: Comparisons of the efficiency (*i.e.*, FLOPs) and the performance (*e.g.*, Acc, mIoU, AP) on four computer vision tasks, *i.e.*, image classification (ImageNet), semantic segmentation (CityScapes), 3D detection (KITTI), and video recognition (HMDB51) between the proposed approach and existing methods. Each method is represented by a circle, whose size represents the number of parameters. ★ represents the optimal model with both high performance and low FLOPs. Our approach achieves superior performance compared to its counterparts on all four tasks.

## C.3 EXPERIMENTS

In this work, we formulate each architecture in NAS-Bench-201 to a $6 * 5 = 30$-dimensional one-hot encoding. We then use our continuous network propagation strategy to traverse architectures in the space with higher-Acc and lower-FLOPs as optimization goals. Given an initial one-hot encoding, our NCP treats it as a continuous-valued coding and utilizes the argmax operation to obtain an edited one-hot coding after every several iterations.

In the experiment, we use the argmax operation after every 10 iterations, termed as a step. As shown in Fig. 6, our NCP finds the optimal structure from a low-Acc and high-FLOPs starting point with only 6 steps (60 iterations). The entire search process takes less than 10 seconds. Compared to other neural predictor-based methods such as (Wen et al., 2020; Luo et al., 2020) that need to predict the accuracy of a large number of architectures, NCP is more efficient.

From Fig. 6 we observe that from the second to fourth steps, the performance of the searched model vibrates. This may be because of the instability caused by one-hot encoding. NCP finds an optimal model with an accuracy of 46.8 and FLOPs of only 90.36 (the highest accuracy in NAS-Bench-201 is 47.33%), showing its effectiveness and generalizability.

## D NCP ON DIFFERENT SINGLE TASKS

We also use NCP to find the optimal architecture on the four basic vision tasks, *i.e.*, image classification, semantic segmentation, 3D detection, and video recognition. Quantitative comparisons in Fig. 7 show that the architectures found by NCP outperform well-designed networks, such as HRNet (Wang et al., 2020a) and ResNet (He et al., 2016).

We visualize the searched architectures by NCP in Fig. 8 to show it can customize different representations for different tasks (objectives). We see that:

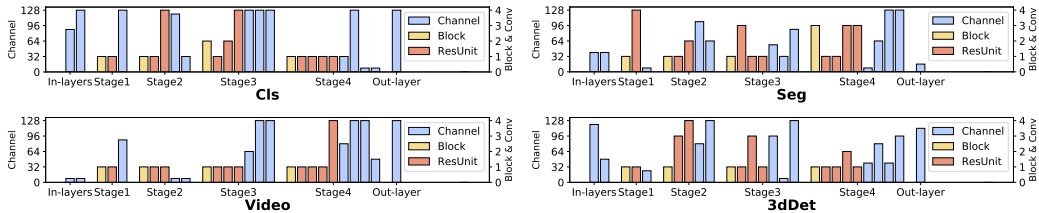

Figure 8: Visualization of the searched models by NCP for four different tasks ($\lambda = 0.5$). The 27-dimensional array in each row represents a network structure.

---

**Algorithm 1** The network propagation process.

1. Learn a neural predictor $\mathcal{F}_W(\cdot)$ and fix it;
2. Initialize an architecture code $e$;
3. Set the target metrics, such as $t_{acc}$ and $t_{flops}$;
**for** each iteration **do**
    4. Re-set new target metrics (optional);
    5. Forward $\mathcal{F}_W(e)$ and calculate the loss;
    6. Back-propagate and calculate the gradient $\Delta e$;
    7. Select an optimal dimension $l$ (optional);
    8. update $e$ based on the gradient $\Delta e$;
**end for**
9. Round and decode $e$ to obtain the final architecture.

---

(1) Segmentation requires the most high-level information in the last two branches of the last stage, and the video recognition model contains the least low-level semantics in the first two stages.

(2) The classification model contains more channels than other tasks, because the FLOPs constraint for classification is the weakest, *i.e.*, classification costs fewer FLOPs than segmentation under the same architecture.

(3) The 3D detection model mainly utilizes the first two branches, which means it may rely more on high-resolution representations.

## E    DETAILS OF NCP

In this section, we provide supplementary details and complexity analysis of our NCP. The algorithm of Network Coding Propagation (NCP) is shown in Algorithm 1. Code is available.

### E.1    IMPLEMENTATION DETAILS

In this work, for each task, 2000 and 500 structures in the benchmark are used as the training set and validation set to train the neural predictor. The optimization goal is set to higher performance (main evaluation metric of each task) and lower FLOPs.

Intuitively, large models can achieve moderate performance on multiple tasks by stacking redundant structures. On the contrary, the lightweight model tends to pay more attention to task-specific structural design due to limited computing resources, making it more suitable for generalizability evaluation. To this end, we set $\lambda = 0.5$ to obtain lightweight models unless specified.

We employ a three-layer fully connected network with a dropout ratio of 0.5 as the neural predictor. The first layer is a shared layer for all metrics with a dimension of 256. The second layer is a separated layer for each metric with a dimension of 128. Then for each evaluation metric, a 128-by-1 fully-connected layer is used for final regression. During training, we use an Adam optimizer with a weight decay of 1e-7. The initial learning rate is set to 0.01 with batch size 128 on a single GPU for 200 epochs, and decays by the one cycle scheduler. The metric prediction is learned using the smoothL1 loss. In the network propagation process, unless specified, we use continuous propagation with an initial code of $\{b, n = 2; c, i, o = 64\}$ and learning rate of 3 for 70 iterations in all experiments.

Table 10: Searching time of predictor-based searching methods. Our NCP is the fastest as it evaluates all dimensions of the code with only one back-propagation without top-K ranking. The time is measured on a Tesla V100 GPU.

| Model | Searching Time | Notes |
|---|---|---|
| Neural Predictor Wen et al. (2020) | > 1 GPU day | Predict the validation metric of 10,000 random architectures and train the top-10 models for final evaluation. |
| NetAdapt Yang et al. (2018) | 5min | Traverse each dimension of the code, predict its per-formance, and then edit the dimension with the highest accuracy improvement. |
| NCP (Ours) | 10s (70 iterations) | Maximize the evaluation metrics along the gradient directions by propagating architectures in the search space. |

## E.2 Time Complexity

Tab. 10 shows the searching time of three predictor-based searching methods. NCP is the fastest as it traverses all dimensions of the code with only one back-propagation without the top-K ranking.

Suffering from the random noise in random search and the neural predictor, existing methods (Wen et al., 2020; Luo et al., 2020) often need to train the top-K models for final evaluation, which is costly (*e.g.*, training a segmentation model on Cityscapes costs 7 hours on 8 Tesla V100 GPUs). NCP uses gradient directions as guidance to alleviate the randomness issue.

## F Visualization and Analysis

### F.1 Analysis of Task Transferring

We visualize the network coding propagation process of our cross-task architecture transferring, *e.g.*, an architecture transferring from the classification task to the segmentation task by using the optimal coding on classification as initialization and using the neural predictor trained on segmentation for optimization. The detailed transferring visualizations of every two of the four tasks are shown in Fig. 9-12 with many interesting findings. For example, all networks in segmentation try to increase the number of channels of the 4th branch of stage 4, while the networks in classification try to decrease it; the networks in video recognition keep it at an intermediate value.

### F.2 Analysis of Single- and Multi-task Searching

We visualize the searched architectures (Fig. 13-14) and the network propagation process of each architecture (Fig. 19-24) in Tab. 8-9.

**Classification.** Manually-designed classification models often use gradually decreasing resolution and gradually increasing the number of channels, because intuitively the learning of classification task requires low-resolution high-level semantic information. However, under different data scales and number of classes, the situation is different, as shown in Fig. 15-18. For example, by reducing the training samples from 1000 to 100 (from Fig. 15 to Fig. 16), the number of convolutional channels and the number of residual units in the 1st branch of stage 4 are increased during propagation, showing the high-resolution low-level information is important when the training samples are insufficient.

**Segmentation.** From Fig 19, Fig 20, and Fig 21 we observe the differences of the network editing process of segmentation models. For example, compared to the two models optimized at a single resolution, the model optimized at both input resolutions reduces the numbers of residual units of the first two stages and increases the number of blocks of stage 2, resulting in more fusion modules and fewer parallel units.

**Segmentation and Video Recognition.** From the visualization of found architecture codes in Fig. 13 and 14, we can find, it is not always that the larger the model, the better the performance. Different objectives result in different customized architecture codes. For example, segmentation tends to select architectures with fewer input channels, while video recognition architectures often have more output channels.

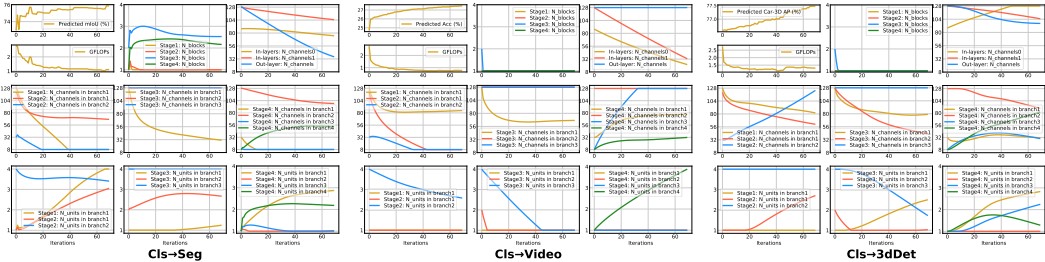

Figure 9: Visualization of our network propagation process of the optimal model in **classification** transferring to other three tasks ($\lambda = 0.5$).

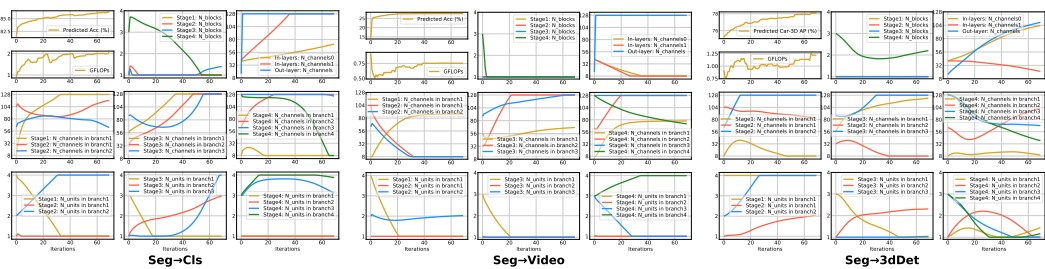

Figure 10: Visualization of our network propagation process of the optimal model in **segmentation** transferring to other three tasks ($\lambda = 0.5$).

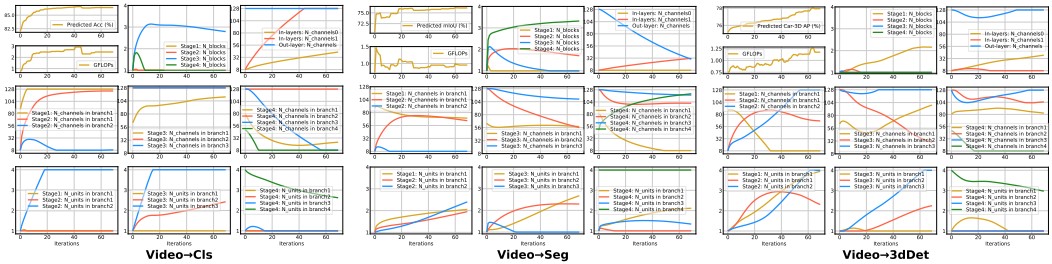

Figure 11: Visualization of our network propagation process of the optimal model in **video action recognition** transferring to other three tasks ($\lambda = 0.5$).

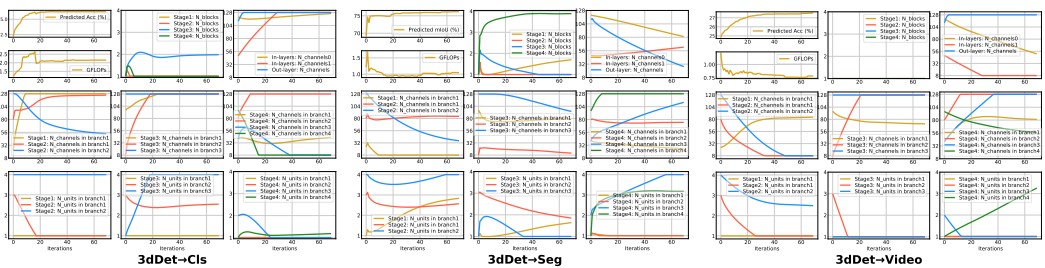

Figure 12: Visualization of our network propagation process of the optimal model in **3D object detection** transferring to other three tasks ($\lambda = 0.5$).

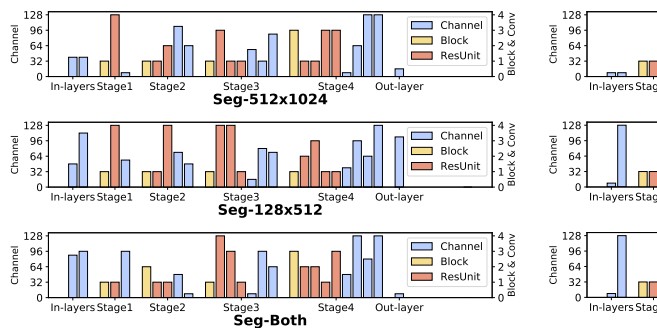
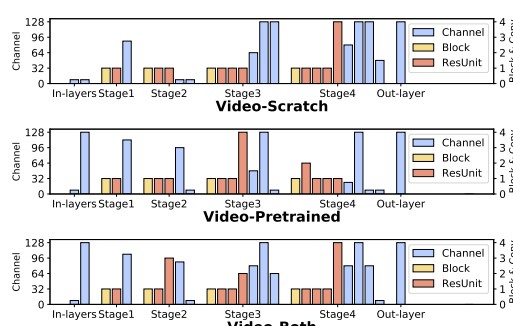

Figure 13: Visualization of the searched **segmentation** models in Tab. 8 by our NCP for intra-task generalizability ($\lambda = 0.5$). The 27-dimensional array in each row represents a network structure.

Figure 14: Visualization of the searched **video recognition** models in Tab. 9 by our NCP for intra-task generalizability ($\lambda = 0.5$). The 27-dimensional array in each row represents a network structure.

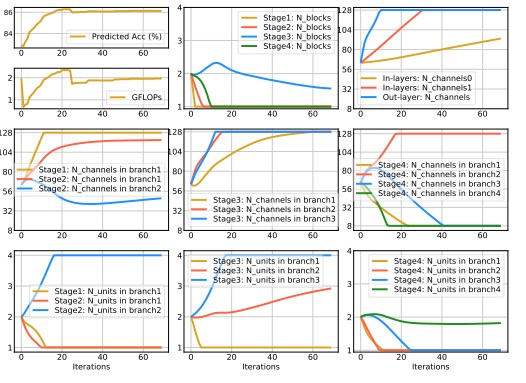
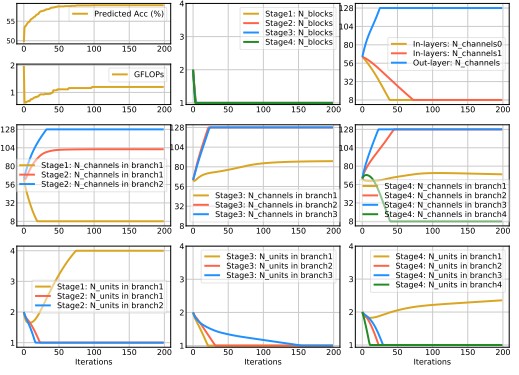

Figure 15: Visualization of our network propagation process of "NCP-Net-A" ($\lambda = 0.7$) for classification.

Figure 16: Visualization of our network propagation process of "NCP-Net-B" ($\lambda = 0.7$) for classification.

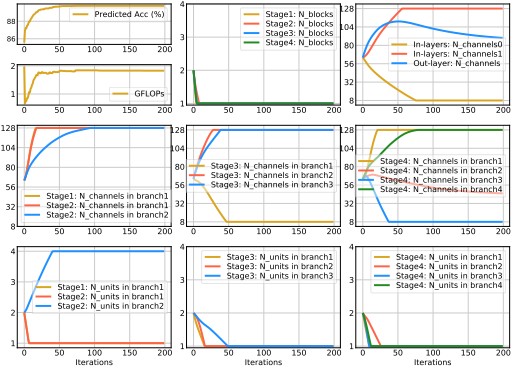
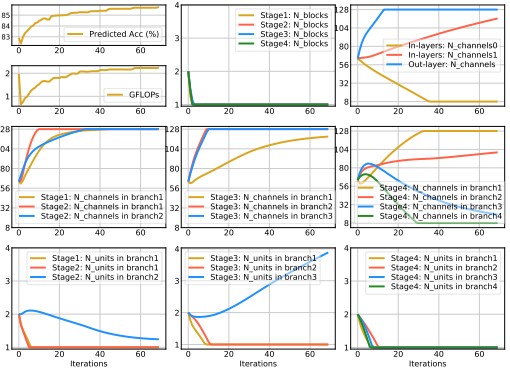

Figure 17: Visualization of our network propagation process of "NCP-Net-C" ($\lambda = 0.7$) for classification.

Figure 18: Visualization of our network propagation process of "NCP-Net-ABC" ($\lambda = 0.7$) for classification.

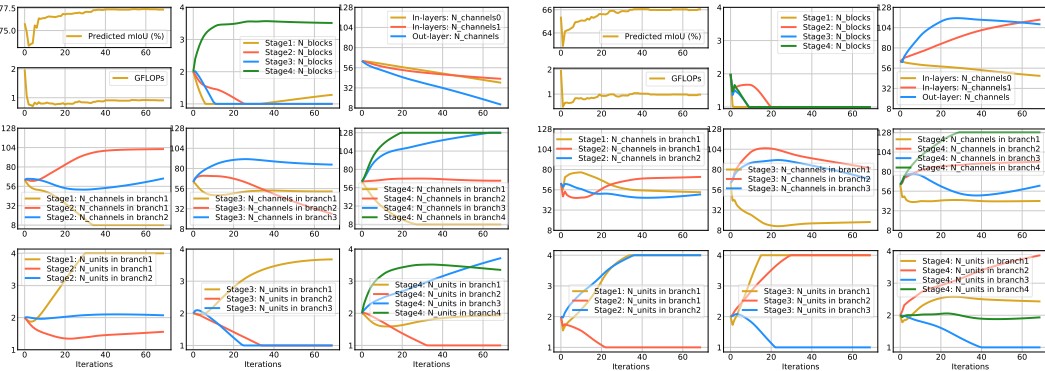

Figure 19: Visualization of our network propagation process of "NCP-Net-512 × 1024" in Tab. 8 (λ = 0.5).

Figure 20: Visualization of our network propagation process of "NCP-Net-128 × 512" (λ = 0.5) in Tab. 8 for segmentation.

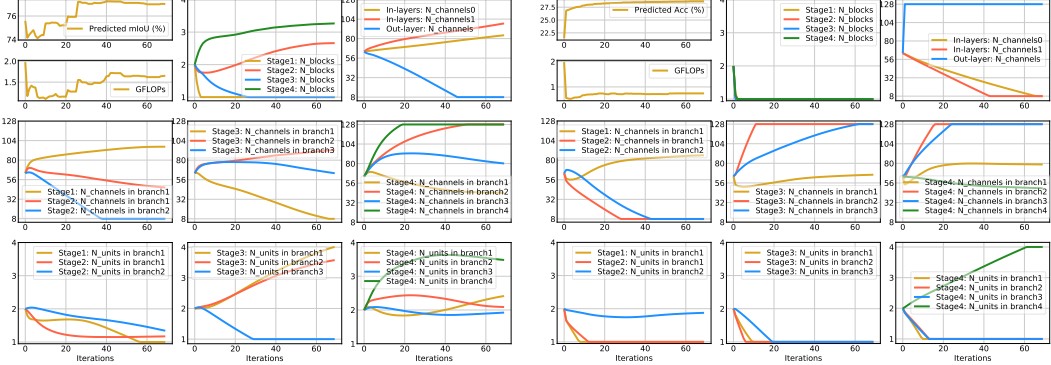

Figure 21: Visualization of our network propagation process of "NCP-Net-Both" (λ = 0.5) in Tab. 8 for segmentation.

Figure 22: Visualization of our network propagation process of "NCP-Net-Scratch" in Tab. 9 (λ = 0.5).

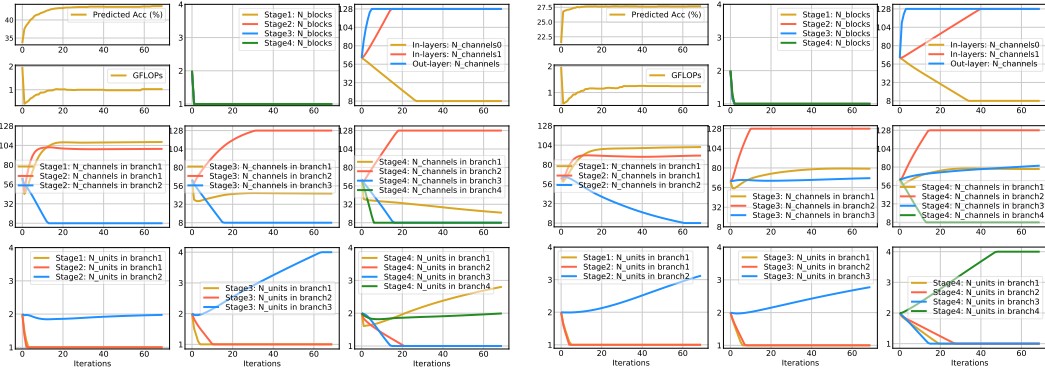

Figure 23: Visualization of our network propagation process of "NCP-Net-Pretrained" (λ = 0.5) in Tab. 9 for video recognition.

Figure 24: Visualization of our network propagation process of "NCP-Net-Both" (λ = 0.5) in Tab. 9 for video recognition.

