# OpenReview forum: "Learning Versatile Neural Architectures by Propagating Network Codes"
_ICLR.cc/2022/Conference — ICLR 2022 Poster_

### Official Review · Reviewer_Z6eY · 2021-11-01

**Correctness:** 4
**Technical Novelty And Significance:** 3
**Empirical Novelty And Significance:** 3
**Recommendation:** 6
**Confidence:** 4

**Main Review:**

The authors first reorganized a NAS multi-task benchmark and trained and evaluate more than 20,000 models on this benchmark. The performance of these models can greatly inspire other following works regarding NAS tasks.

For the proposed NCP method, the overall idea is simple and sound. The overall assumption of the NCP is that the search space for model code e is continuously differentiable. Then the whole problem is essentially an optimization problem: given several samples [e_i, y_i], we wanted to find the \hat{e} so that can maximize y. The authors trained a three-layer MLP as a predictor (function: f: e-->y). Here I feel curious and wish to raise a question, though what I can think about is also training an MLP as a predictor and using gradient descent to find the optimal, if some classic convex optimization algorithms can also solve this problem? And how about the performance? It is okay not to find another solution.

Another question is about the generalization of this method. If we would like to apply NCP for a new benchmark, e.g. COCO for segmentation, how can we use the knowledge we have on NAS-Bench-MR to find a proper parameter and architecture for the model? Since for NAS-Bench-MR, the authors trained a large number of the models to have the search space, which is not generalizable for each new dataset.

Besides, the writing of this paper needs to be further revised. The current version is a little hard to follow and needs re-read to get the whole idea of the authors.

Some typos with a glance:
1) abstract: wildly used --> widely used
2) caption of Fig. : 3th --> 3rd
-----------------------------
After rebuttal:
Thanks for answering my questions in detail and the rebuttal fully addressed my concerns. I think this paper can be accepted to ICLR 2022.

**Summary Of The Paper:**

This paper organized a multi-task NAS benchmark including 4 widely used datasets and more than 20,000 models. It also proposed a searching architecture named Network Coding Propagation (NCP) to effectively find the optical model for specific tasks. Experimental results indicate that the proposed model can be applied to inter-task, cross-task, and intra-tasks problems and the authors also showed the generalization capability to other benchmarks.

**Summary Of The Review:**

The NAS benchmark (NAS-Bench-MR) contains valuable large-scale experimental results and be used to benefit other NAS research. The proposed NCP method is sound but kind of naive. The writing of this paper needs to be further revised for both method's details and organizations. Considering all these conditions, I would suggest accepting this paper to ICLR but would not champion it.

---

> ### Author Response · Authors · 2021-11-20
> **Response to Reviewer #Z6eY**
>
> Thank you for the insightful comments and suggestions.
>
> **Q1. [If some classic convex optimization algorithms can also solve this problem? It is okay not to find another solution.]** \
> Thanks for the suggestion. We have compared our work with some hyperparameter optimization methods, such as NetAdapt [A] and Random Search [B], as shown in Tab. 2 of our paper.
>
> Besides, towards similar motivation, we could use Bayesian optimization (BO) [C, D] to perform automatic architecture design. Using BO, an architecture’s performance is typically modeled as sampling from a Gaussian process (GP). We will try to implement it shortly for performance comparison.
>
> [A] Yang, Tien-Ju, et al. "Netadapt: Platform-aware neural network adaptation for mobile applications." Proceedings of the European Conference on Computer Vision (ECCV). 2018. \
> [B] Bergstra, James, and Yoshua Bengio. "Random search for hyper-parameter optimization." Journal of machine learning research 13.2 (2012). \
> [C] Snoek, Jasper, Hugo Larochelle, and Ryan P. Adams. "Practical bayesian optimization of machine learning algorithms." Advances in neural information processing systems 25 (2012). \
> [D] Kandasamy, Kirthevasan, et al. "Neural architecture search with bayesian optimization and optimal transport." arXiv preprint arXiv:1802.07191 (2018).
>
>
> **Q2. [Generalization. E.g., apply NCP for a new benchmark, COCO for segmentation.]** \
> Good question. In Table 3 of our paper, we demonstrate the generalizability of this work by applying the searched model to a new segmentation dataset, ADE20K, which is not used to train neural predictors. Our models show good performance and generalizability. Therefore, a simple way is to directly use the optimal model searched on the Cityscapes segmentation task for the COCO dataset. The model with only 360 GFLOPs (measured using 800x1280 resolution, Mask R-CNN head) achieves 47.2\% bounding box AP and 42.3\% mask AP on the detection and instance segmentation tasks, respectively. With fewer computational costs, our results outperform HRNet-W48 (46.1\% for bbox AP and 41.0\% for mask AP) by a large margin.
>
> In addition, we can use the models in our benchmark as pre-training models, and finetune some architectures on the COCO dataset as a benchmark for the instance segmentation task. And the neural predictor can also be obtained by finetuning the Cityscapes segmentation predictor, which significantly reduces the computation required to build a new benchmark. We have tried this finetuning strategy on video recognition, as shown in Table 6 in Appendix.
>
>
> **Q3. [The writing of this paper needs to be further revised.]** \
> Thanks. We will revise our paper carefully to make it clearer and easy to read.
>
> We have incorporated some of your comments and suggestions in the newly uploaded version. And we will also do so for the rest in the next revision. Thanks again for your time and effort!

---

### Official Review · Reviewer_BCgR · 2021-11-02

**Correctness:** 4
**Technical Novelty And Significance:** 4
**Empirical Novelty And Significance:** 3
**Recommendation:** 6
**Confidence:** 5

**Main Review:**

strength:
1. Good idea. The NCP trained on network codes instead of original data. Therefore the trained network can be used to update the architecture efficiently across datasets.
2. Extensive experiments are conducted. Experiments are conducted on three settings: inter-tasks, cross-tasks, and intra-tasks. Results on these datasets demonstrate the effectiveness of the proposed NCP.

weakness:
1. The main weakness is about the evaluation of the main idea, network coding spaces. The main contribution of NCP is that NCP converts them into network coding spaces. However, very little understanding or intuition about the network coding space is provided. For example, is it continuous? Can we do interpolation between two codes? It is hard to tell what are the benefits and what are the disadvantages of optimizing on network coding spaces. Properties about the network coding spaces need to be further elaborated.
2. The paper NAO (Luo et al., 2020) also propose to search on a continuous space. The difference between the proposed method and previous NAO needs more discussion. Why the proposed updating architecture codes along desired gradient directions for various objectives is novel, compared to NAO?



**Summary Of The Paper:**

Network Coding Propagation is proposed in the paper for NAS on multiple heterogeneous vision tasks.


**Summary Of The Review:**

The idea and the experiments of the paper are good, but some components need more explanation. Also, comparison with existing work needs to be more specific.

---

> ### Author Response · Authors · 2021-11-20
> **Response to Reviewer #BCgR**
>
> Thank you for the insightful comments and suggestions.
>
> **Q1. [Properties about the network coding spaces need to be further elaborated. Is network coding space continuous? Can we do interpolation between two codes?]** \
> Good question. Many previous works have explored the use of predictors in the neural network hyper-parameter space. For example, [A] first proposes to encode a given architecture to continuous embedding space via RNN. [B, D] then builds the accuracy predictor based on LSTM/GIN to perform gradient-based
> search. Further, [F] points out that [A, B] are easy to overfit and challenging to apply to different tasks, and [C, E, F] propose to directly optimize in the network hyper-parameter space using GCNs and GBDTs. These successful experiences show that the hyperparameter space of the network is approximately "continuous" that can be optimized directly.
>
> We have taken another step forward from the previous works. We define a more "continuous" space and use a gradient-based MLP neural predictor. Considering that the design space used by the earlier work was not continuous enough as they converted a list of operator choices into one-hot embedding, we make our coding space contain only natural values (1,2,3,4,...) instead of one-hot embeddings. Specifically, we define the number of layers, blocks, and channels on four branches to build a continuous-valued coding space which makes interpolation (integer) between two codes possible. In this way, our space can be approximately continuous. More detailed discussions can be found in Sec. 3.2 (coding) and Appendix C.2.
>
> [A] Baker, Bowen, et al. "Accelerating neural architecture search using performance prediction." ICLR 2018. \
> [B] Luo, Renqian, et al. "Neural architecture optimization." NeurIPS 2018. \
> [C] Wen, Wei, et al. "Neural predictor for neural architecture search." ECCV 2020. \
> [D] Yan, Shen, et al. "Does unsupervised architecture representation learning help neural architecture search?." NeurIPS 2020. \
> [E] Dudziak, Łukasz, et al. "Brp-nas: Prediction-based nas using gcns." NeurIPS 2020. \
> [F] Luo, Renqian, et al. "Accuracy Prediction with Non-neural Model for Neural Architecture Search." arXiv preprint arXiv:2007.04785 (2020).
>
>
> **Q2. [The difference between the proposed method and previous NAO [B] needs more discussion.]** \
> Method [B] builds the accuracy predictor based on LSTM to perform a gradient-based search. The differences between our work and [B] lie in:
> - Different design space, the space of [B] contains discrete one-hot embeddings modeled by LSTM encoders and decoders. On the contrary, we define a continuous-valued space where an MLP predictor can directly work.
> - [B] jointly trains an encoder, a performance predictor, and a decoder to minimize the combination of performance prediction loss and structure reconstruction loss. We learn and inverse the neural predictor directly in our coding space.
> - [B] relies on the implicit gradient by updating the embedding (output from the encoder). Thus its network editing is non-transparent and discontinuous in a gradient update step (an update may cause a dramatic architecture change). On the contrary, our gradient update and network editing are explicit, guiding us to understand and design better structures by showing how the structure is improved step by step (in one or several dimensions at a time).
>
> [B] Luo, Renqian, et al. "Neural architecture optimization." NeurIPS 2018.
>
> We have incorporated some of your comments and suggestions in the newly uploaded version. And we will also do so for the rest in the next revision. Thanks again for your time and effort!

---

### Official Review · Reviewer_AfiZ · 2021-11-07

**Correctness:** 4
**Technical Novelty And Significance:** 3
**Empirical Novelty And Significance:** 4
**Recommendation:** 8
**Confidence:** 2

**Main Review:**

**Pros**


* This paper tackles a relevant problem of finding versatile neural network architectures that should be suitable for a multitude of vision-related tasks that require different granularities of learned representations. This could bring the community a step closer towards unifying vision tasks and tackling several vision tasks (detection, segmentation …) using a single neural network. This has a potential for significant impact for the community: having general models and backbones could significantly ease the deployment of perception models in real-world scenarios (having several networks trained independently and tackling different sub-problems, and, in turn, requiring multiple forward passes, is not really a practical solution).
* The problem is well-stated, and the methodology seems sound to me. To study this (cross-task) NAS problem, the paper first formalizes a test-bed, needed to study this problem, defines a search space that seeks representations that extract features at different granularities, and finally, proposes a method that can traverse the search space efficiently (and is applicable across tasks).
The paper is well-written and reasonably accessible to a reader, completely unfamiliar with NAS (i.e., this reviewer).
* The experimental evaluation is thorough and supports the main claims (i.e., the proposed method can find both specialized network architectures as well as architectures that perform very well on recognition and segmentation tasks).


**Cons**

* From the eye of someone that is not familiar with NAS methodology and terminology, I was rather confused with a brief description of what the proposed neural coding propagation does (high-level description in the intro). I could only fully understand this part after fully reading Sec. 3.
From the related work, it was unclear whether there have been any similar efforts in other fields (such as NLP) of formalizing NAS for finding versatile (instead of specialized) network architectures?
* I think it would be fair to make it more explicit that the search space of the network architectures this method can find is still rather constrained and limited to convolutional neural networks and well-established and tested design patterns. It appears to me that the method will not be able to find a radically novel architecture but rather find “good configurations” within a well-constrained space.
* The paper states that not all candidate models need to be evaluated by the neural predictor, but only those that are “in the search space along the gradient direction”. I would expect a more thorough justification for this. Also, how is exactly the search space along the gradient direction defined?


**Summary Of The Paper:**

This paper tackles neural architecture search (NAS), addressing the problem of finding architectures suitable for a multitude of vision-related tasks, ranging from object detection to semantic segmentation. To the best of my knowledge, this is the first attempt in NAS for computer vision models that explicitly design the search space that allows for a search of architectures that are broadly applicable to a variety of tasks that usually rely on the different granularity of feature representation.
This paper makes two important steps towards the automated search of such general, multi-purpose architectures: (1) to study this problem, the paper formalizes a multi-task NAS benchmark, covering recognition, detection, semantic segmentation, and activity recognition (video) datasets. This allows for assessing the generality of the sought architecture across vision tasks. (2) to tackle the architecture search in such a setting, this paper propose makes the following contributions. First, it augments the search space with multiple stages that extract feature maps at several resolutions, followed by a feature fusion step (the number of blocks and feat. map resolutions are governed by the network hyperparameters.). This seems to be one of the key features that help to find architectures suitable for tasks that differ as much as image classification and semantic segmentation.
Next, for assessing the performance in this large search space, this paper builds on predictor-based methods, where the idea is to train a network that predicts the performance (i.e., the learned predictor) of the sought-network based on the (encoded) hyperparameters, governing the network architecture. Similarly, the proposed neural coding propagation (NCP) directly encodes the hyperparameters in the network coding space — those can thus be directly updated via back-propagation of the learned predictor. A predictor is learned for each task independently, and this way, gradients can be accumulated separately across different tasks before jointly updating the “architecture codes”.

**Summary Of The Review:**

I am absolutely no expert in the NAS area, so I can only provide an assessment of a reader that is not at all familiar with the related work. However, I find it that this paper tackles a very important problem of general cross-task vision architecture search and makes important steps in this direction by providing an evaluation test-bed and methodology that experimentally validates the claims. In particular, Tab. 2 confirms that the proposed method finds several well-performing network architectures for semantic segmentation, while ‘Cls+seg’ entry in Tab. 3 works very well for both semantic segmentation and ImageNet classification. I see this as evidence that the proposed methodology is indeed capable of finding versatile neural network architectures.

Based on what I can conclude based on the manuscript I see this paper as a very solid contribution, and I think this paper should be accepted. I hope, however, that the other reviewers will be more familiar with NAS and can provide deeper insights (and spot potential issues that were missed by an “untrained eye,” in case they are present).

** After rebuttal **

Thanks for answering my questions in the rebuttal! All my doubts were clarified. I have read other reviews and didn't spot any major weaknesses. I am retaining my rating (accept).

---

> ### Author Response · Authors · 2021-11-20
> **Response to Reviewer #AfiZ**
>
> Thank you for the positive comments and insightful suggestions.
>
> **Q1. [The high-level description in the introduction is confusing.]** \
> Sorry for the confusion. We will revise the introduction and show more details about our NCP methodology for a clearer understanding.
>
>
> **Q2. [Whether there have been any similar efforts in NLP?]** \
> Thank you for the suggestion. We studied recent NAS works on NLP and found some related work, but there is no work similar to ours. We summarize them as follows.
>
> A few previous studies [C, D] have attempted to optimize existing conventional recurrent cells.  They concluded that the ordinary LSTM performs reasonably well on all datasets, and no modification improves its performance significantly. [B] further applies NAS to search for a better alternative to the Transformer. However, their searched architectures are only tested on the language translation task. Another relevant work to ours is [A], which constructs a NAS-Bench-NLP benchmark with 14k trained architectures in search space of recurrent neural networks on two language modeling datasets. However, they focus more on the benchmark/dataset itself rather than a network architecture search method for versatile architectures.
>
> [A] Klyuchnikov, Nikita, et al. "NAS-Bench-NLP: neural architecture search benchmark for natural language processing." arXiv preprint arXiv:2006.07116 (2020). \
> [B] So, David, Quoc Le, and Chen Liang. "The evolved transformer." International Conference on Machine Learning. PMLR, 2019. \
> [C] Greff, Klaus, et al. "LSTM: A search space odyssey." IEEE transactions on neural networks and learning systems 28.10 (2016): 2222-2232. \
> [D] Jozefowicz, Rafal, Wojciech Zaremba, and Ilya Sutskever. "An empirical exploration of recurrent network architectures." International conference on machine learning. PMLR, 2015.
>
>
> **Q3. [Make it more explicit that the search space is limited to CNNs and well-established and tested design patterns.]** \
> Thanks for the suggestion. We will make it more clearer and explicit.
>
> All previous NAS methods find optimal architectures in given search spaces, as you said, "find good configurations within a well-constrained space". Our NCP follows this paradigm. Nobly, unlike many existing search strategies that work on a specific search space, our NCP is widely applicable to other spaces, such as NAS-bench-201 [E], and achieves good performance (see Sec.C of the appendix).
>
> [E] Dong, Xuanyi, and Yi Yang. "Nas-bench-201: Extending the scope of reproducible neural architecture search." ICLR 2020.
>
> **Q4. [A thorough justification for "not all candidate models need to be evaluated by the neural predictor".]** \
> In this work, we represent our network search space as a continuous-valued coding space, rather than one-hot embedding (choices of different operators) in previous works [F, G]. In this way, our space can be seen as a continuous space, making gradient-based optimization possible. Then we get the gradients by inverting the neural predictor and applying them to the network coding to update the architecture. In this way, only the architectures in the gradient direction are evaluated, and we can find optimal models in several back-propagations.
>
> [F] Wen, Wei, et al. "Neural predictor for neural architecture search." ECCV 2020. \
> [G] Dudziak, Łukasz, et al. "Brp-nas: Prediction-based nas using gcns." NeurIPS 2020.
>
> We have incorporated some of your comments and suggestions in the newly uploaded version. And we will also do so for the rest in the next revision. Thanks again for your time and effort!

---

### Author Response · Authors · 2021-11-20
**General Response**

We sincerely appreciate all reviewers’ time and efforts in reviewing our paper. We are glad to find that reviewers generally recognized our contributions:
* **Method.** Bring the community a step closer towards unifying vision tasks and tackling several vision tasks using a single neural network [AfiZ]; Have potential for significant impact for the community [AfiZ]; Good and sound idea [BCgR, Z6eY]; Inspire other following works regarding NAS tasks [Z6eY].
* **Experiment.** Experimental evaluation is thorough and supports the main claims [AfiZ, BCgR].
* **Writing.** The problem is well-stated and the paper is well-written [AfiZ].

We also thank all reviewers for their insightful and constructive suggestions, which help further improve our paper. We have incorporated some of their comments and uploaded a new version. In addition to the pointwise responses below, we summarize the supporting experiments/statements added in the rebuttal according to reviewers’ suggestions.
* Summarized related works in NLP [AfiZ];
* Made the introduction of our method more explicit and clearer [AfiZ, Z6eY];
* Detailed our continuous-valued coding space and the difference with previous work (NAO) [BCgR];
* Extended our searched model to COCO instance segmentation [Z6eY].

We hope our pointwise responses below could clarify all reviewers’ confusion. We thank all reviewers’ time again. If you have any questions, please feel free to let us know. We appreciate your suggestions and comments!

---

### Author Response · Authors · 2021-11-28
**Summary of our rebuttal and discussion**

We sincerely appreciate all reviewers’ and ACs’ time and efforts in reviewing our paper. We truly thank you all for the insightful and constructive suggestions, which helped further improve our paper. We genuinely appreciate the positive 8-6-6 evaluation from reviewers AfiZ, BCgR, and Z6eY.

As the deadline for discussion is approaching, we are happy to provide any additional clarifications that you may need. In our previous response, we have carefully studied your comments and made detailed responses summarized below:

* [Additional Experiments] As suggested by reviewer Z6eY, we conducted extra experiments by extending our searched model to COCO instance segmentation. The results consistently validate the effectiveness of our proposal.
* [Clarifications] We detailed our continuous-valued coding space and made further comparisons with NAO (Luo et al., 2020) and more related works in NLP.
* [Writing] We owe many thanks to reviewers AfiZ and Z6eY’s extremely helpful writing suggestions. We have incorporated some comments in the revision. All improved manuscript parts, together with other constructive discussions with reviewer BCgR, will be delivered in our final version.

We really thank all reviewers’ and ACs’ time and efforts again. If you have any more questions, please feel free to let us know.

Best, \
Authors

---

### Decision · Program_Chairs · 2022-01-20

**Decision:**

Accept (Poster)

**Comment:**

The paper examines neural architecture search for multi-task networks, by associating model hyperparameters with a coding space and building an MLP predictor for mapping codes to task performance.  After the discussion phase, reviewers are marginally in favor or accepting the paper, pointing to the extensive experimental results as a convincing contribution.